# Repurposing Caffeine, Metformin, and Furosemide to Target Schizophrenia-Related Impairments in a Triple-Hit Rat Model

**DOI:** 10.3390/ijms26136019

**Published:** 2025-06-23

**Authors:** Gyongyi Horvath, Szonja Bianka Plesz, Eszter Ducza, Dorottya Varga, Edina Szucs, Sándor Benyhe, Leatitia Gabriella Adlan, Gabor Braunitzer, Gabriella Kekesi

**Affiliations:** 1Department of Physiology, Albert Szent-Györgyi Medical School, University of Szeged, Dóm tér 10, H-6720 Szeged, Hungary; plesz.szonja.bianka@med.u-szeged.hu (S.B.P.); adlan.leatitia@med.u-szeged.hu (L.G.A.); kekesi.gabriella@med.u-szeged.hu (G.K.); 2Department of Pharmacodynamics and Biopharmacy, Faculty of Pharmacy, University of Szeged, H-6720 Szeged, Hungary; ducza.eszter@szte.hu (E.D.); vdorottya0731@gmail.com (D.V.); 3Delta Bio2000 Kft., H-6726 Szeged, Hungary; szucs.edina@deltabio.eu; 4Institute of Biochemistry, HUN-REN Biological Research Centre, H-6726 Szeged, Hungary; benyhe.sandor@brc.mta.hu; 5Sztárai Institute, University of Tokaj, H-3950 Sárospatak, Hungary; braunitzergabor@gmail.com

**Keywords:** Long Evans, schizophrenia, caffeine, furosemide, metformin, behavior, D2 dopamine receptor, Ambitus

## Abstract

The limited efficacy of antipsychotics in treating the negative and cognitive symptoms of schizophrenia has prompted the exploration of adjuvant therapies. Several drugs developed for other indications—including caffeine, metformin, and furosemide—have shown procognitive potential. This study evaluated the effects of these agents on behavioral parameters using the reward-based Ambitus test, and on the cerebral D2 dopamine receptor (D2R) expression and binding. The drugs were administered individually and in combination in a schizophrenia-like triple-hit animal model (Lisket rats), derived from the Long Evans (LE) strain. Lisket rats received 14 days of drug treatment via drinking water; water-drinking LE rats served as the controls. The Ambitus test was conducted before treatment and on days 11–14. Caffeine enhanced activity without affecting learning or memory. Metformin and furosemide reduced exploratory behavior but improved reference memory; these effects were inhibited by caffeine co-administration. Although no statistically significant behavioral differences were found compared to water-treated Lisket rats, a trend toward reduced exploratory visits was observed in the triple-combination group. Lisket rats exhibited moderately reduced D2R binding in the cortex and increased binding in the hippocampus. Caffeine alone and in combination enhanced hippocampal D2R binding, while furosemide increased cortical D2R expression. This study is the first to highlight the behavioral and molecular effects of these non-antipsychotic agents in a schizophrenia model, supporting their potential for adjunctive use.

## 1. Introduction

Schizophrenia is a complex neuropsychiatric disorder characterized by positive, negative, and cognitive symptoms [1,2,3]. Unfortunately, antipsychotics are largely ineffective in treating negative symptoms and cognitive deficits [4]. Therefore, various adjuvant therapies have been proposed as complementary treatments [5,6,7]. Drug repurposing is a strategic approach aimed at achieving effective therapy by applying widely used drugs to conditions other than their original indication. This method reduces both the high costs and lengthy timelines associated with the development of new drugs for clinical use [8].

Caffeine, a naturally occurring methylxanthine and a widely consumed psychoactive substance, is frequently used by patients with schizophrenia [9]. It exerts stimulant effects—such as reduced drowsiness and fatigue, and enhanced locomotor activity—by acting as a non-selective adenosine A1/A2A receptor antagonist [10,11,12]. Caffeine may also modulate negative symptoms of schizophrenia and reduce the extrapyramidal and sedative side effects of antipsychotics [13,14,15]. However, studies have produced conflicting evidence regarding its influence on learning processes under various conditions, including in patients with schizophrenia and animal models [10,15,16,17,18,19,20,21,22,23,24,25,26].

Psychiatric conditions are often associated with an elevated risk of metabolic syndrome, including hyperglycemia and hypertension, typically treated with the antidiabetic drug metformin and the loop diuretic furosemide, respectively [6,27,28]. Metformin has been shown to improve cognition in Parkinson’s and Alzheimer’s diseases [29,30], and some studies have explored its behavioral effects in rodent models of schizophrenia [31,32,33]. Furosemide, which targets sodium–potassium–chloride cotransporter 1 (NKCC1) and potassium–chloride cotransporter 2 (KCC2), significantly influences GABAergic neurotransmission [34]. Diuretics have demonstrated beneficial effects in a range of CNS disorders, including epilepsy, anxiety disorders, autism spectrum disorder, and, to a lesser extent, schizophrenia [35,36,37,38,39,40,41,42,43].

Translational research relies on animal models that replicate key features of human diseases. Given the critical role of gene–environment interactions in schizophrenia, complex multiple-hit models offer valuable insights. The “Lisket” rat, derived from the Long Evans strain, was developed through a combination of post-weaning social isolation, subchronic ketamine (an NMDA receptor antagonist) treatment, and selective breeding for behavioral traits—following the same paradigm used for the Wisket model (based on the Wistar strain) [44,45,46]. Long Evans (LE) rats demonstrate greater activity and cognitive performance than Wistar animals [47,48,49], and have been used as a single-hit schizophrenia model [50,51]. The Lisket model was designed to enhance face validity for schizophrenia-like behaviors, while avoiding the marked inactivity observed in Wistar and Wisket animals [52,53].

Dopamine dysfunction in schizophrenia affects several brain regions, notably the cerebral cortex (CTX) and hippocampus, and includes decreased dopamine levels in these areas accompanied by increased D2 receptor density [54,55,56,57,58]. We previously reported altered D2 receptor (D2R) binding and expression in Wisket rats, modulated by caffeine treatment [18,59]. However, the impact of metformin and furosemide on D2R function in schizophrenia models remains unexplored.

The first aim of this study was to assess the effects of subchronic (14-day) oral treatment with caffeine, metformin, furosemide, and their double and triple combinations on parameters related to locomotor and exploratory activity, as well as cognitive functions, in Lisket rats using a reward-based learning test (Ambitus; Figure 1A). The second aim was to evaluate D2R binding and protein expression in the CTX and hippocampus of water-drinking control and Lisket animals, and to examine the impact of the aforementioned treatments in Lisket rats. To this end, one control group (LE_W; water only) and eight Lisket groups (based on drug treatment) were included in a 14-day protocol (days 1–14). Lisket groups received water (Lisket_W), caffeine (CAFF, 0.5 mg/mL), metformin (MTF, 1 mg/mL), furosemide (FURO, 0.1 mg/mL), or combinations (CAFF_MTF, CAFF_FURO, MTF_FURO, and TRIPLE; Figure 1B). This study may enhance the face and predictive validity of this new triple-hit rat model and provide insights into the efficacy of commonly used drugs for improving the cognitive and behavioral symptoms of schizophrenia.

## 2. Results

### 2.1. General Observations

The body weight of the Lisket animals was significantly lower throughout the entire experiment (Group: F (1, 174) = 14.22; *p* < 0.001), with no significant differences observed among the various drug-treated Lisket groups during treatment (F (7, 150) = 0.26; *p* = 0.97). Water consumption was similar between the LE and Lisket animals. However, fluid intake among the Lisket groups was significantly influenced by pharmacological treatment (F (7, 74) = 3.11; *p* < 0.01). Post hoc analysis revealed that seven groups (CAFF, MTF, FURO, CAFF_MTF, MTF_FURO, TRIPLE) did not differ significantly from one another, consuming a similar daily volume (mean: 104 ± 3.0 mL/kg/day). In contrast, the CAFF_FURO-treated group (146 ± 11.5 mL/kg/day) drank significantly more fluid than the other groups, except for the TRIPLE-treated group (126 ± 13.7 mL/kg/day). In the CAFF_FURO group, the intake of caffeine and furosemide was significantly higher (73 ± 5.8 mg/kg and 15 ± 1.2 mg/kg, respectively) compared to their respective monotherapy groups (53 ± 1.9 mg/kg and 10 ± 0.5 mg/kg, respectively). The higher drug dose in this group likely resulted from increased fluid consumption, presumably due to compensatory drinking following diuresis.

### 2.2. Behavioral Differences Between the Water-Drinking LE and Lisket Animals

Repeated-measures ANOVA revealed significant effects of group or group-by-phase interaction for most parameters, except for the latency of the first non-baited box visit (Lat_NRW) and reference memory (R_M). However, none of the z-scores showed significant effects for phase alone (Table 1). These results indicate that Lisket animals exhibited several behavioral impairments (Figure 2). Post hoc analysis showed that exploratory activity on the rewarded side and impairments in learning capacity were observed in nearly all phases, while for other parameters, only a few phases showed significant differences between the two groups. For example, significant group differences in locomotion and working memory were observed only at the final phase.

### 2.3. Behavioral Results in Pharmacological Treatments in Lisket Animals

One-way ANOVA revealed significant treatment effects for several parameters (Expl, Expl_FR_NRW, Lat_RW, Lat_NRW, skipping, R_M; as indicated in Table 1 and Figure 3). Caffeine monotherapy caused a moderate but non-significant increase in exploratory behavior, accompanied by a significantly shorter latency to the first visit to a non-baited box (Lat_NRW) compared to the water-drinking Lisket group (Lisket_W). Although baseline R_M did not differ between Lisket_W and LE_W, both metformin and furosemide monotherapies significantly enhanced reference memory values compared to the water- and caffeine-treated groups. These improvements were accompanied by a moderate reduction in overall exploration. Only furosemide significantly increased the Lat_NRW and also led to significant attentional impairment, as indicated by elevated skipping behavior compared to Lisket_W animals.

With regard to the CAFF_MTF combination, caffeine co-administration blunted both the metformin-induced hypoactivity and the enhancement in reference memory. The CAFF_FURO combination resulted in an extremely high level of exploratory activity toward both rewarded and non-baited boxes, with short latencies, which was accompanied by impaired reference memory compared to furosemide alone. Surprisingly, the MTF_FURO and TRIPLE combinations produced attenuated effects compared to the individual drugs, with most values resembling those of the water-drinking Lisket group—suggesting some form of antagonism between the ligands.

### 2.4. In Vitro Results

#### Receptor Binding Assay

The maximal D2R binding capacities (B_max_) were lower in the cerebral cortex (CTX) compared to the hippocampus across all groups (Figure 4A; Table 2). Regarding differences between the water-drinking LE and Lisket animals, D2R B_max_ was moderately increased in Lisket animals in the hippocampus (*p* = 0.12), while the opposite trend was observed in the CTX (*p* = 0.2). Caffeine treatment, both alone and in combination, led to elevated B_max_ values in both brain regions, reaching statistical significance in the hippocampus when compared to the LE_W, Lisket_W, MTF, and FURO groups, while metformin and furosemide did not influence this parameter significantly. The overall pattern of treatment effects appeared similar in the CTX and hippocampus. Although the trends in dissociation constant (K_d_) values paralleled those of B_max_, the differences did not reach statistical significance for this parameter.

### 2.5. Receptor Protein Expression

In the CTX, a tendency toward increased D2R protein expression was observed in the water-drinking Lisket animals compared to the LE group (*p* = 0.06), which was reversed by caffeine treatment (Figure 4B). Furosemide alone, as well as in combination with caffeine, showed a strong tendency toward enhanced receptor protein expression, reaching statistical significance when compared to the caffeine monotherapy group. No significant differences in D2R protein expression were detected between groups in the hippocampus.

## 3. Discussion

This study characterized the behavioral phenotype and cerebral D2R alterations of a triple-hit schizophrenia rat model derived from the Long Evans (LE) strain and, for the first time, presented the effects of subchronic treatment with caffeine, metformin, and furosemide—both as monotherapies and in combination—on these parameters. Lisket rats exhibited behavioral impairments in the Ambitus test at both young (10th week) and adult (13th week) ages, accompanied by moderate changes in D2R binding and expression in the hippocampus and CTX. None of the treatments had a significant effect on the acquisition learning capacity (L_C), a parameter based on the number of rewards consumed and the eating time, which may reflect the animals’ eagerness to obtain rewards. However, caffeine treatment alone enhanced activity-related parameters and increased D2R binding and was associated with a tendency toward decreased reference memory (Figure 3 and Figure 4). Metformin and furosemide moderately reduced exploratory activity while improving reference memory. Furthermore, furosemide also caused significant impairments in the latency to visit non-rewarded boxes and in attentional performance. Decreased reference memory—calculated as the ratio of explorations into the rewarded boxes to total explorations—may reflect long-term memory impairments, although this measure can also be influenced by elevated exploratory activity.

Notably, furosemide also increased D2R protein expression in the CTX. Combinations of caffeine with either metformin or furosemide—similar to caffeine alone—enhanced behavioral activity and D2R binding, primarily in the hippocampus. Additionally, the CAFF_FURO combination moderately increased D2R protein expression in the CTX.

Cognitive dysfunction remains an unresolved issue in schizophrenia and is closely linked to impaired behavioral activity [60,61,62,63,64]. Our earlier results showed that Wisket animals exhibited significant alterations in locomotion, exploration, and learning ability [44,46], whereas Lisket animals presented with mild motor deficits but severe cognitive impairment—suggesting that cognitive decline in these animals was not attributable to changes in activity levels. Multiple brain regions (including the hippocampus and CTX), which are involved in cognitive functions and operate through several neurotransmitters, interact to form the complex pathophysiology of schizophrenia [65,66,67]. A decreased cortical density of D2Rs has been demonstrated in patients with schizophrenia [68], although some studies reported no differences compared to healthy controls prior to antipsychotic treatment [69]. Data from schizophrenia animal models are limited and show inconsistent findings. For example, increased D2R protein expression has been reported in the prefrontal cortex (PFC) in a double-hit mouse model of schizophrenia [70]. In our earlier study, we observed elevated B_max_ values in the CTX and hippocampus, along with a reduction in the PFC, in Wisket animals [59]. A similar trend in hippocampal B_max_ enhancement was found in the Lisket animals, but with an opposite pattern in the CTX. This discrepancy may be due to methodological differences: in the earlier study, CTX and PFC samples from Wisket animals were measured separately, whereas in the current study, cortical samples from Lisket animals were not dissected. Strain differences between Wistar and LE rats may also play a role, as LE rats are known to exhibit higher levels of activity and cognitive function than Wistar animals [47,48,49]. Given the significant roles of both the hippocampus and CTX in cognitive processes, we propose that even moderate changes in D2R binding and protein expression may be associated with the behavioral alterations observed in Lisket animals.

As current antipsychotics fail to fully address cognitive and behavioral impairments—and often cause significant side effects—adjuvant therapies are being actively investigated for schizophrenia. In this study, caffeine, metformin, furosemide, and their combinations influenced several behavioral and D2R-related parameters in a triple-hit rat model. Notably, patients with schizophrenia frequently use these agents, often concurrently—for example, consuming coffee and/or taking metformin or furosemide to manage comorbidities. Therefore, evaluating their combined effects in a schizophrenia model is both timely and clinically relevant.

Caffeine is a natural methylxanthine with psychostimulant properties, acting as an adenosine receptor antagonist. The adenosinergic system—including adenosine and its G-protein-coupled receptors (A1, A2A, A2B, and A3)—is involved in regulating several behavioral processes, including motivation, cognitive function, and motor activity [11]. Increased coffee consumption among patients with schizophrenia is well documented, although its effects remain controversial [14,71]. While caffeine may exacerbate psychotic symptoms and worsen schizophrenia [14], improved learning and memory have also been reported in both animal and human studies [72,73,74,75]. However, other studies report that caffeine has no effect on memory [76,77,78] or even impairs it [21,22,23]. Preclinical studies have likewise produced inconsistent behavioral findings regarding caffeine treatment [10,18,19,20,71,73,79]. Our earlier findings in Wisket rats demonstrated that a 5-day caffeine injection produced an immediate increase in behavioral activity, followed by delayed hypoactivity and reduced learning ability [18]. In contrast, in the present study, caffeine increased activity in Lisket animals without significantly improving learning capacity. These differences may be due, at least in part, to variations in drug administration methods (voluntary drinking vs. injection) and strain differences.

Nevertheless, these findings support previous suggestions that caffeine treatment could serve as a potential pharmacological approach for addressing motivational impairments in schizophrenia [63,77]. The interpretation of behavior in caffeine-treated groups must take into account its concomitant motor-stimulant effect [11,80,81], as heightened exploratory activity may have contributed to the observed decrease in reference memory performance. In the pathophysiology of neuropsychiatric disorders, the modulatory role of adenosine on various neurotransmitter systems may be highly relevant [11,72,81,82,83]. Increased basal D2R occupancy observed in patients with schizophrenia may reduce the influence of A2A receptors (A2ARs) on D2Rs, thereby increasing the receptors’ affinity for dopamine. Our earlier findings in Wistar and Wisket animals revealed elevated B_max_ values in the CTX and hippocampus of Wisket rats, alongside reduced values in the PFC of the Wistar group. In both strains, caffeine treatment significantly increased D2R protein expression in the hippocampus [18]. In the present study, caffeine similarly enhanced B_max_ values in Lisket animals, primarily in the hippocampus; however, no change in receptor protein expression was observed in this region. The slight discrepancies between our earlier and current findings highlight the potential impact of strain differences and variations in the experimental paradigm.

Metformin (dimethyl-biguanide), a derivative of the traditional herbal medicine *Galega officinalis*, is widely prescribed for type 2 diabetes [84]. Its repurposing for neuropsychiatric disorders has been proposed, given its potential to reduce brain disturbances linked to insulin resistance, oxidative stress, and neuroinflammation [85]. Metformin crosses the blood–brain barrier and has been shown to improve cognitive function and reduce anxiety [29,30,86,87]. A few studies have also reported its beneficial behavioral effects in schizophrenia models [31,32,33]. Our recent findings showed that metformin did not affect learning capacity in Wisket rats but reduced clozapine-induced behavioral impairments [31]. Interestingly, Lisket rats appeared more sensitive, showing improved reference memory during treatment. The mechanisms behind metformin’s effects are incompletely understood. Both AMP-activated protein kinase (AMPK)-dependent and -independent pathways have been implicated [88,89]. Metabolic dysregulation is common in brain disorders, and AMPK dysfunction may be linked to schizophrenia [90]. A recent study showed that metformin’s anxiolytic effects involve AMPK-dependent action on GABAergic cortical interneurons [87]. Only one in vitro study has reported metformin-induced upregulation of D2R mRNA and protein expression in pituitary prolactinoma cells [91], which contrasts with our in vivo findings, as no significant D2R-related changes were observed in Lisket rats.

Furosemide is one of the most frequently used diuretics, prescribed for various conditions such as hypertension. Interestingly, loop diuretics have been shown to exert anxiolytic effects by attenuating stress-induced behavioral and hormonal changes [35,37,92], and to improve learning and memory deficits in models of Huntington’s and Down syndromes [93,94]. Limited evidence also suggests that these drugs may alleviate cognitive disturbances and/or positive symptoms of schizophrenia [36,41,42]. In the present study, furosemide, like metformin, significantly improved reference memory, and caused a moderate reduction in behavioral activity. Loop diuretics act through complex mechanisms, including the inhibition of NKCC1 and KCC2 channels with equal potency [34], the modulation of alpha6 subunit-containing GABA_A_ receptors [95], and a reduction in extracellular sodium and potassium levels [96]. The overall effect of these mechanisms likely depends on the expression levels of the respective channels. NKCC1 mediates chloride accumulation into neurons, leading to excitatory GABA_A_ receptor activity, while KCC2 promotes chloride extrusion, resulting in inhibitory GABAergic action. The combined activity of these transporters results in low intracellular chloride concentrations in healthy adult neurons [97], and the activation of GABA_A_ receptors typically causes hyperpolarization via chloride influx, leading to synaptic inhibition. When the expression of these transporters is disrupted, neuronal inhibition via GABA_A_ receptor activation can fail, contributing to various neurological disorders, including epilepsy, autism spectrum disorder, and schizophrenia [38,98,99,100]. These conditions are typically associated with increased NKCC1 expression and decreased KCC2 expression, or at least an elevated NKCC1/KCC2 ratio. Given that abnormalities in the GABAergic system are well documented in patients with schizophrenia, the modulation of NKCC1 may help restore altered inhibitory function in the brain [38]. Although we did not assess NKCC1 expression in this study, the observed positive effects of furosemide on memory function may suggest partial restoration of the balance between these two transporters in our model. To date, no data are available on the effects of loop diuretics on D2R function. In our study, furosemide treatment did not significantly alter B_max_ values in the examined brain regions; however, it did increase D2R protein expression in the CTX. The discrepancy between results from the two in vitro assays may be attributable to methodological differences between radioligand binding and Western blot techniques and/or to an increased receptor number accompanied by decreased binding affinity. The exact implications of the enhanced cortical D2R protein expression in the effects of furosemide remain unclear, as the CAFF_FURO combination produced similar changes in receptor expression but was associated with different behavioral outcomes. It cannot be excluded that the functionality of the D2R receptors is differentially influenced by these ligands; however, further in vitro studies are required to explore the role of furosemide in D2R functionality.

Regarding the drug combinations, caffeine co-administered with metformin or furosemide reversed the effects of these compounds on exploratory activity and reference memory. These antagonistic behavioral interactions were accompanied by increased D2R binding in both brain regions without changes in receptor protein expression. A possible explanation for the discrepancies between the binding and D2R protein expression may be that enhanced receptor expression is accompanied by decreased ligand binding affinity, or that altered binding occurs despite normal protein expression. Furthermore, the binding capacity values in radioligand binding assays do not always parallel those in protein expression studies performed by Western blot analysis, as they can be influenced by various, partly experimental factors, such as receptor affinity, receptor occupancy, ionic environment and regulation, the presence or absence of guanine nucleotides, allosteric interactions, internalization, or trafficking. These results suggest that D2R affinity for its ligand may be enhanced by these treatments, depending at least on part on the dopamine levels. The elevated exploratory activity in the CAFF_FURO group may reflect higher drug intake due to increased fluid consumption, likely compensating for the enhanced diuretic effect [101]. Surprisingly, the behavioral outcomes of the MTF_FURO and TRIPLE combinations resembled those of the water-drinking group, further supporting antagonistic interactions between the different drugs. Although combination therapy complicates pharmacological interpretation, both pharmacokinetic and pharmacodynamic interactions may be involved. In terms of pharmacokinetics, antagonism might occur at the intestinal level (influencing absorption), during metabolism, or in transport across the blood–brain barrier. In addition, altered bioavailability due to enhanced diuresis and differences in fluid consumption may have contributed to significant variations in drug effects. Pharmacodynamic interactions between caffeine and the other two ligands might result from their opposing effects on brain activity; caffeine, as a psychostimulant, enhances it by enhancing dopamine and acetylcholine release [102], whereas metformin and furosemide may act by enhancing GABAergic inhibition [87,97]. However, the exact mechanisms underlying these effects cannot be determined from the current study, and further in vitro and in vivo investigations are needed.

Overall, the findings suggest that non-antipsychotic agents such as metformin, furosemide, and caffeine may have potential relevance in the treatment of schizophrenia, warranting further investigation into their mechanisms and applications.

### Limitations

Our findings should be interpreted with caution. First, only Lisket animals received drug treatments, as the aim was to assess effects within the triple-hit schizophrenia model; this approach also aligned with the 3R principle by reducing the number of animals used. Second, drugs were administered via drinking water to enhance translational relevance, though this method introduced variability due to difference in individual fluid intake. Daily gavage, while more precise, was avoided to prevent repeated stress. Lastly, animals were single-housed throughout the experiment. Although this may have induced some stress, it allowed for accurate monitoring of individual fluid and food consumption.

## 4. Methods

### 4.1. Animals

Male Long Evans (LE; control; n = 6) and Lisket (n = 83) rats were included in this study. All experiments were conducted with the approval of the Hungarian Ethical Committee for Animal Research (registration numbers: XIV/1248/2018 and XIV/1421/2023) and in accordance with the guidelines established by the Government of Hungary and EU Directive 2010/63/EU for animal experimentation. Although no specific humane endpoints were reached during the study, a predefined exclusion criterion was established: animals would have been excluded from the experiment if their body weight had decreased markedly (e.g., by 15%) due to excessive diuretic effects. Animals were maintained on a 12 h light/dark cycle under controlled conditions of temperature (22 ± 1 °C) and humidity (55 ± 10%) conditions. The body weight of the rats was monitored weekly throughout the entire experiment.

The number of animals in each group was chosen to allow comparison with our previous work [31], and was as follows: LE_Water: 6; Lisket Water: 9; and 10 animals each in the caffeine (CAFF; Sigma Aldrich, Budapest, Hungary), metformin (MTF; Meforal, Berlin-Chemie Menarini, Budapest, Hungary) and caffeine–furosemide (CAFF_Furo)-treated groups. Eleven animals were included in the furosemide (FURO; Furon, Teva, Debrecen, Hungary), CAFF_MTF, MTF_FURO, and TRIPLE treatment groups. A larger number of animals was used in the drug-treated groups to account for expected variability due to individual differences in drug sensitivity. All drugs were dissolved in tap water and administered via the drinking bottle.

### 4.2. Experimental Paradigm

As in our earlier studies [18,103], Lisket animals were housed individually after weaning (at 4 weeks of age) for 28 days and received intraperitoneal ketamine treatment (30 mg/kg/day for 5 days) during the second week of isolation. They were then re-housed in groups (2–3 per cage) and allowed a one-week recovery period without further interventions. Resocialization did not lead to aggressiveness or injury in these animals. Control animals were socially housed throughout and did not receive ketamine or vehicle injections to prevent them from any stress. Baseline behavioral testing was conducted at 10 weeks of age using the Ambitus apparatus without any prior training (Ambitus1; Figure 1B). After 48 h of food restriction (no food with water ad libitum), two tasks were performed: Task 1 (Trials 1–2; all boxes baited; PRE_ALL phase) in the morning, and Task 2 (Trials 3–4; only internal boxes baited; PRE_IN phase) three hours later (Figure 1B). Thus, four trials were conducted using Ambitus1 test on a single day. The intertrial interval was 3 h between morning and afternoon sessions and 2 min between trials within each session. At least one week later, at the age of three months, all rats were single-housed during the drug administration period to allow for accurate monitoring of individual fluid consumption. One control group (LE_W; water only) and eight Lisket groups (based on drug treatment) were included in a 14-day protocol (days 1–14). Lisket groups received water (Lisket_W), caffeine (CAFF, 0.5 mg/mL), metformin (MTF, 1 mg/mL), furosemide (FURO, 0.1 mg/mL), or combinations (CAFF_MTF, CAFF_FURO, MTF_FURO, and TRIPLE; Figure 1B). Drug doses were based on previous rodent studies that showed effectiveness and were approximately equivalent to human therapeutic levels [30,31,104,105]. Animals were assigned to groups using stratified randomization based on baseline behavioral scores and body weight to ensure comparability. A second Ambitus test (Ambitus 2) was conducted on days 11–14, following two days of food deprivation (no food, water ad libitum on days 9–10), with four trials per day. Testing phases were defined by the task used: POST_ALL (Task 1, Trials 1–2, day 11), POST_EX (Task 3, Trials 3–4 on day 11 and Trials 5–8 on day 12), and POST_IN (Task 2, Trials 9–16 on days 13–14). In total, 16 post-treatment trials were conducted over four days, using the same intertrial interval as in Ambitus1. Moderate food restriction was maintained throughout testing (about 25 g/kg/day), and drinking fluid was available ad libitum. Body weight was recorded weekly, and total fluid (and drug) intake, normalized to body weight, was calculated. Behavioral experiments were conducted between 8:00 AM and 4:00 PM under dim lighting.

On day 15, the animals were decapitated, and their brains were rapidly removed. The CTX and hippocampus were immediately dissected, frozen in liquid nitrogen, and stored at –75 °C until used in in vitro assays. Based on behavioral results, the MTF_FURO and TRIPLE groups were excluded from in vitro analysis.

### 4.3. Ambitus Test

The Ambitus apparatus (from Latin ambitus, meaning “going around”) is a reward-based cognitive test apparatus, which consists of a rectangular corridor constructed from clear Plexiglas on a black floor (Deák Delta, Ltd., Mártély, Hungary, Figure 1A) [44]. Each corridor contains 4 side boxes—2 on the inner and 2 on the outer sides—for a total of 16 boxes, used to deliver food rewards (puffed rice, 20 mg). At the beginning of each trial, rewards were placed in the boxes, and the rat was positioned at the starting point (Figure 1A), after which the experimenter immediately exited the room. Animals were allowed to explore the corridor and collect food rewards for 300 s. The apparatus was cleaned between animals using 70% ethanol to eliminate olfactory cues. Infrared beams detected exploratory behavior, reward collection at each side box, and locomotor activity at the midpoint of each corridor, with a time resolution of 1 millisecond. The Ambitus system enabled the calculation of multiple parameters, including locomotor, exploratory, and eating activities, as well as cognition-related measures such as effective exploration (E_E), attention (measured by skipping), learning capacity (L_C), working memory (W_M, reflecting short-term memory), and reference memory (R_M, reflecting long-term memory), which are defined and calculated as detailed in Table 1. As the Ambitus is a custom-designed, square-shaped corridor with side boxes, it can detect the attentional deficits by measuring skipping—defined as the number of corridors walked without exploration.

### 4.4. In Vitro Experiments

#### 4.4.1. Preparation of Brain Samples for Receptor Binding Assays

Neuronal membrane fractions were prepared for in vitro receptor binding according to our previous studies [106,107]. The brains were homogenized in 30 volumes (*v*/*w*) of ice-cold 50 mM Tris-HCl buffer (pH 7.4) using a Teflon-glass Braun homogenizer operating at 1500 rpm. The homogenate was centrifuged at 18,000 rpm for 20 min at 4 °C; the resulting supernatant was discarded, and the pellet was resuspended in the original volume of Tris-HCl buffer. The homogenate was then incubated at 37 °C for 30 min in a shaking water-bath. Then centrifugation was repeated as described before. The final pellet was suspended in 5 volumes of 50 mM Tris-HCl pH 7.4 buffer and stored at −80 °C. The protein content of the membrane preparation was determined using the method of Bradford method with BSA as the standard [108].

#### 4.4.2. Receptor Binding Experiment

Receptors, like enzymes, can be saturated with their specific ligands and, therefore, their maximum binding capacities can be measured. In saturation binding assays, radiolabeled ligands are applied at graded concentrations, and specific binding is quantified relative to ligand concentration to determine binding kinetics. Aliquots of frozen rat brain membrane homogenates were thawed, centrifuged, and resuspended in 50 mM Tris-(hydroxymethyl)-aminomethane hydrochloride (Tris-HCl; Sigma-Aldrich, Budapest, Hungary) buffer (pH 7.4). Equilibrium saturation binding assays were performed using the selective D2R antagonist [^3^H]spiperone (specific activity: 80.2 Ci/mmol; PerkinElmer, Boston, MA, USA) at increasing concentrations (0.09 nM–4.95 nM) at 25 °C for 120 min. Ketanserin (1 µM; Thermo Fisher GmbH, Kandel, Germany) was used to block radioligand binding to 5-HT_2_ serotonergic receptors [109]. Non-specific binding was determined using 10 µM unlabeled spiperone (Tocris Bioscience, Bristol, UK). The reaction was terminated by rapid filtration under vacuum (Brandel M24R Cell Harvester; Brandel Inc. Gaithersburg, MD, USA) and washed three times with 5 mL of ice-cold 50 mM Tris-HCl buffer (pH 7.4) through Whatman GF/B glass fibers filters. Radioactivity from the dried filters was measured using an UltimaGold^TM^ MV aqueous scintillation cocktail on a Packard Tricarb 2300TR liquid scintillation counter (PerkinElmer, Boston, MA, USA). The saturation-binding assays were performed in duplicate and were repeated at least three times.

#### 4.4.3. Western Blot Analysis

Brain tissues were homogenized using a Micro-Dismembrator (Sartorius AG, Göttingen, Germany) and centrifuged at 11,000 rpm for 30 min at 4 °C in RIPA Lysis Buffer System (Santa Cruz Biotechnology, Dallas, TX, USA). Total protein content in the supernatant was determined by spectrophotometry (BioSpec-nano, Shimadzu, Kyoto, Japan). A total of 25 µg of protein per well was subjected to electrophoresis on 4–12% NuPAGE Bis-Tris Gel in XCell SureLock Mini-Cell Units (Thermo Fisher Scientific, Budapest, Hungary) at 200 V, 100 mA (start), and 80 mA (run) for 50 min. Proteins were transferred from gels to nitrocellulose membranes, using the iBlot Gel Transfer System (Thermo Fisher Scientific, Budapest, Hungary). Antibody binding was detected with the WesternBreeze Chromogenic Western Blot Immunodetection Kit (Thermo Fisher Scientific, Budapest, Hungary; 20 V, 7 min at room temperature). The membrane blocking was made for 30 min, on a rotary shaker at room temperature with blocking buffer of WesternBreeze Chromogenic Western blot immunodetection kit (Thermo Fisher Scientific, Budapest, Hungary). Blots were then incubated overnight at 4 °C on a shaker with polyclonal antibodies against D2R (Proteintech, Bio-Kasztel Ltd., Budapest, Hungary) and β-actin (bs-10966R, Bioss Antibody, BioTech Hungary Ltd.; Budapest, Hungary) in blocking buffer. Images were captured using the EDAS290 imaging system (Kodak Ltd., Budapest, Hungary), and the optical density of each immunoreactive band was determined with Kodak 1D Image Analysis Software version 4.0. Optical densities were calculated as arbitrary units following local area background subtraction and normalized to β-actin.

### 4.5. Measurements and Statistical Analyses

For clarity of presentation, data obtained from the Ambitus test were normalized as z-scores by subtracting the mean value from each individual score and dividing the result by the standard deviation of the population, calculated separately for the five phases (PRE_ALL, PRE_IN, POST_ALL, POST_EX, and POST_IN; Figure 1B). It is important to note that several parameters could only be calculated for trials in which rewards were placed on one side (Table 1: parameters 4, 6, 10, and 11); therefore, an asymmetric reward configuration was used during the PRE_IN, POST_EX, and POST_IN phases. No primary outcome variable was predefined, as this was an exploratory study aimed at identifying whether any treatment effects could be detected across the measured parameters. Values reflecting enhanced behavioral activity or learning functions were assigned positive z-scores, while for variables where higher raw values indicated impairments (e.g., latency, skipping), z-score were inverted, resulting in negative z-scores (e.g., a high latency value was expressed as a negative z-score).

Repeated-measures ANOVA was applied to compare body weight and Ambitus test results across the five phases between LE_W and Lisket_W animals. The effects of treatments on mean fluid and drug intake, as well as on the mean z-scores during the POST_EX and POST_IN phases, were analyzed using one-way ANOVA, as all behavioral parameters were available in these two phases, providing reliable information about drug-related effects. Table 1 provides definitions and summarizes the statistical significance of the behavioral parameters analyzed. Post hoc comparisons were performed using Fisher’s LSD test. The assumption of sphericity was evaluated for repeated-measures ANOVA, and the Greenhouse–Geisser correction was applied where appropriate to adjust for violations.

Regional and treatment-related changes in dopamine D2 receptor (D2R) function were assessed using [^3^H]spiperone radioligand binding assays on membrane fractions prepared from cortical (CTX) and hippocampal tissue samples of rat brains. Equilibrium saturation binding data were analyzed by nonlinear regression, assuming a single class of binding sites; therefore, the “one site–specific binding” fitting model in GraphPad Prism version 10.2.1 was applied. The maximum binding capacity (Bmax) was expressed in fmol/mg protein, calculated based on total protein content, radioligand concentration, and molar specific activity.

The same number of animals used in the behavioral study was included in the in vitro experiments. For the binding assays, tissue samples from the same brain region were pooled within each treatment group to obtain sufficient material, and measurements were performed in triplicate. For the Western blot analyses, at least six individual samples per group were included, excluding outliers with values beyond the mean ± 2 standard deviations. An unpaired two-tailed *t*-test was used to assess statistical significance in both types of in vitro experiments.

All data are presented as means ± S.E.M., and significance was set at *p* < 0.05. Statistical analyses were performed using STATISTICA 13.4.0.14 (TIBCO Software Inc., Palo Alto, CA, USA) and GraphPad Prism (GraphPad Software Inc., San Diego, CA, USA).

## 5. Conclusions

In conclusion, this study provides the first detailed behavioral and receptor-level assessment of various drug treatments in a chronic, triple-hit schizophrenia model derived from the Long Evans strain. Lisket rats exhibited persistent behavioral deficits and moderate changes in D2R binding—including a decrease in the cerebral cortex and an increase in the hippocampus—some of which were improved by pharmacological intervention. Our findings support the repurposing of commonly used drugs for the adjunctive treatment of schizophrenia, particularly for targeting cognitive and behavioral impairments that are insufficiently managed by antipsychotics. The results also highlight the importance of considering interaction effects in combination treatments targeting both behavioral and receptor-level outcomes.

## Figures and Tables

**Figure 1 ijms-26-06019-f001:**
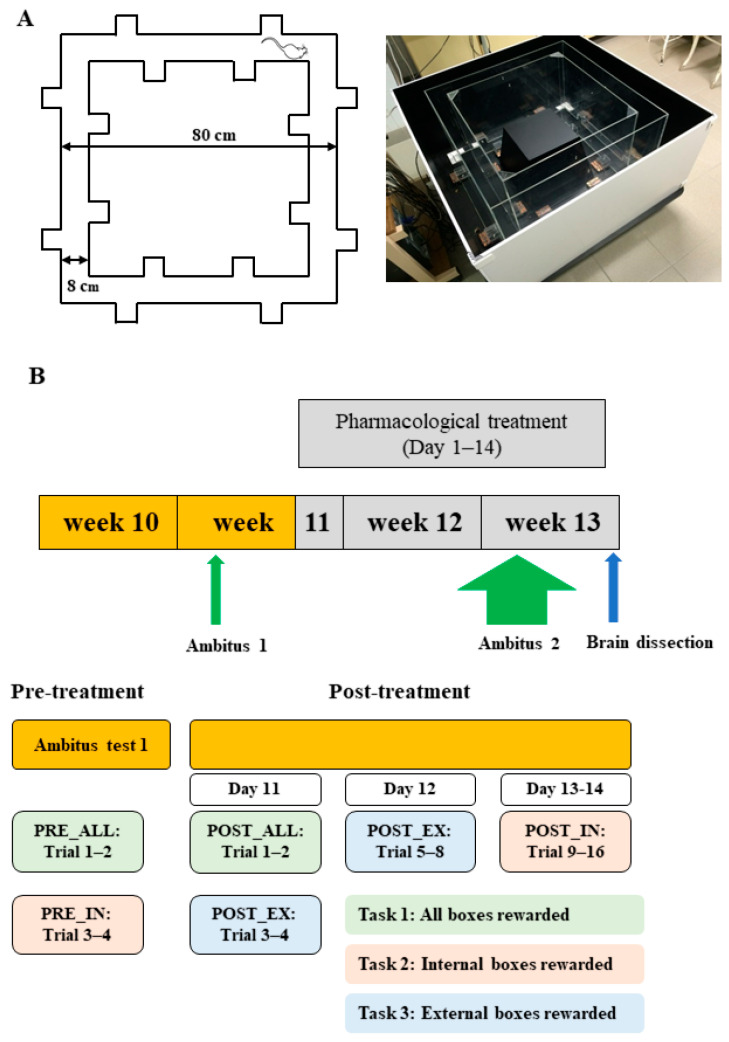
(**A**) The Ambitus apparatus. (**B**) Experimental paradigm. The colors correspond to the specific task applied during each phase (PRE_ALL, PRE_IN, POST_ALL, POST_EX, and POST_IN). Green, pink, and blue indicate Tasks 1, 2, and 3, respectively.

**Figure 2 ijms-26-06019-f002:**
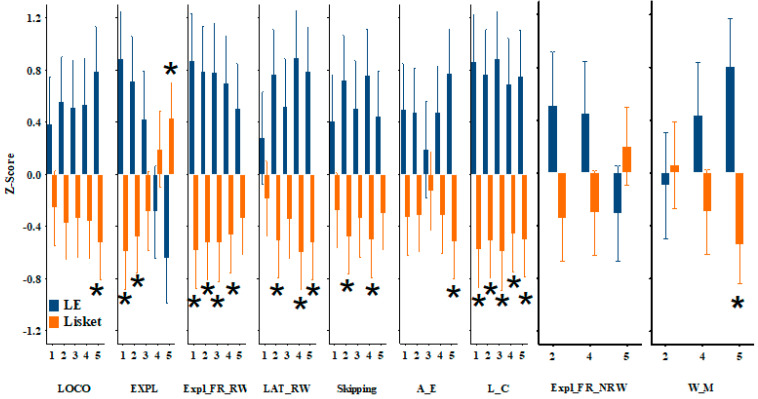
Behavioral parameters showing significant effects of group or group-by-phase interaction (expressed as z-scores) between LE-water and Lisket-water animals across the phases of the Ambitus test. Definitions, abbreviations, calculation methods, and statistical results for these parameters are provided in Table 1. Asterisks (*) indicate significant differences between groups (*p* < 0.05). Note that values indicating enhanced behavioral activity or learning functions have positive z-scores, whereas parameters where higher raw values reflect impairments (e.g., latency, skipping) are shown with negative z-scores (e.g., a high latency value corresponds to a negative z-score).

**Figure 3 ijms-26-06019-f003:**
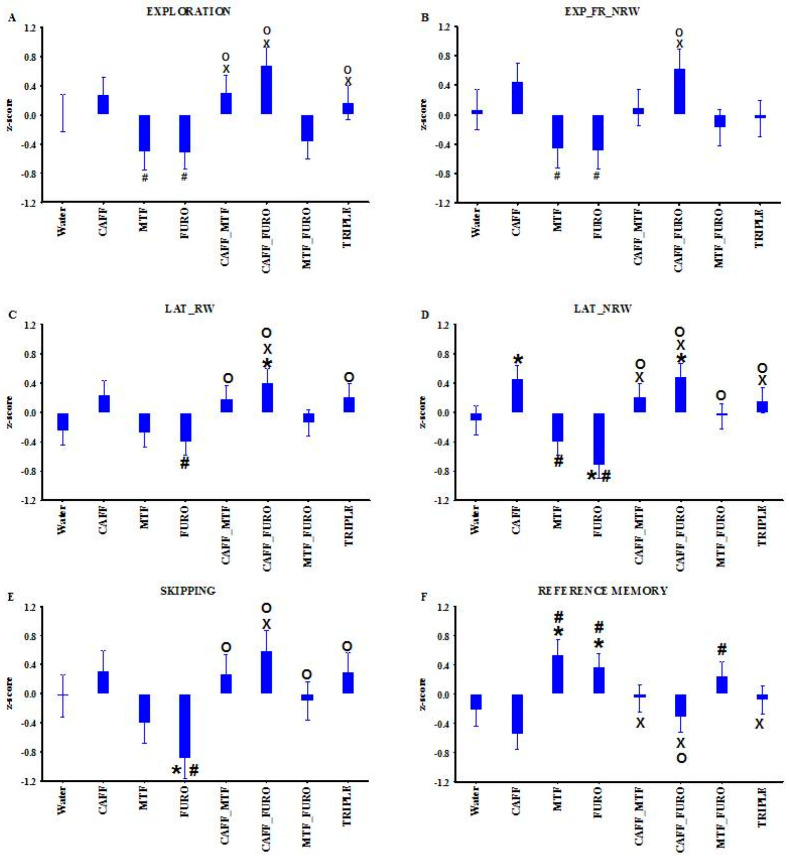
(**A**–**F**) Behavioral parameters showing significant differences (expressed as z-scores) across treatment groups. Definitions, abbreviations, calculation methods, and statistical results for the analyzed parameters are provided in Table 1. Symbols indicate significant differences (*p* < 0.05) compared to the following treatment groups: water (*), caffeine (#), metformin (×), and furosemide (o). Note that z-scores are positive for values indicating enhanced behavioral activity or learning functions, while parameters where high raw values indicate impairments (e.g., latency, skipping) are represented with negative z-scores (e.g., a high latency value is shown as a negative z-score).

**Figure 4 ijms-26-06019-f004:**
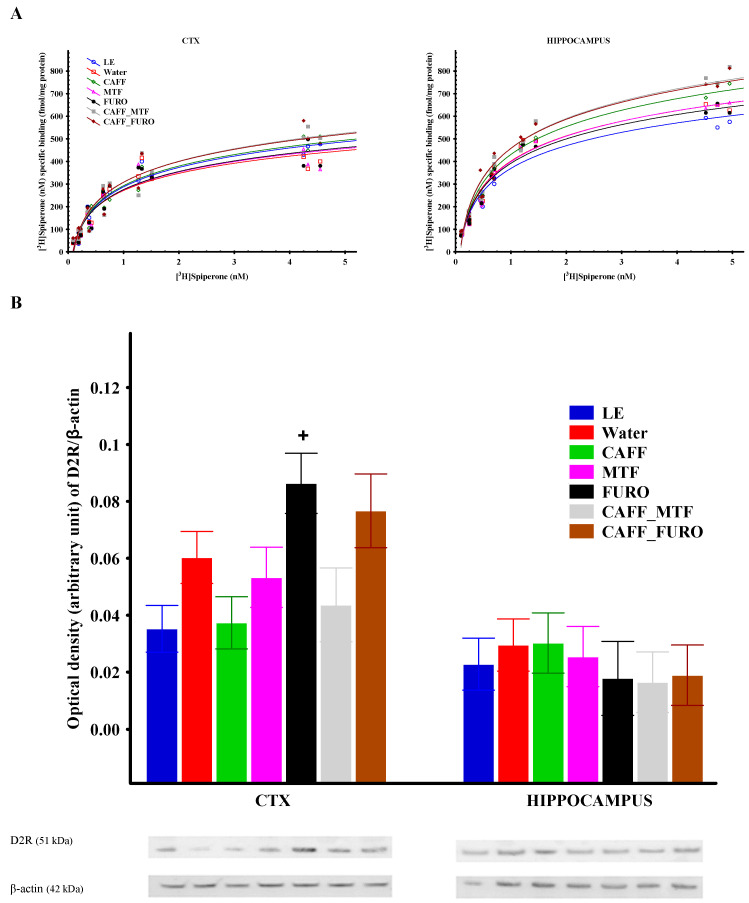
(**A**) Equilibrium saturation isotherms and statistical results of specific [³H]spiperone binding to D2Rs in rat brain membrane homogenates from the cerebral cortex (CTX) and hippocampus. The graphs show the amount of D2R protein (fmol/mg) specifically bound to [³H]spiperone at increasing radioligand concentrations for each group. (**B**) The expression of D2Rs measured by Western blot in the CTX and hippocampus across treatment groups. The symbol (+) indicates a significant difference (*p* < 0.05) compared to the caffeine (+)-treated group.

**Table 1 ijms-26-06019-t001:** Definitions and calculation methods (first column) along with statistical results for behavioral parameters, analyzed by group (second and third columns; LE vs. Lisket; repeated-measures ANOVA) and by pharmacological treatment in Lisket animals (fourth column; one-way ANOVA).

Parameters Definition: Calculation	Group	Group and Phase Interaction	Treatment
1. Locomotion (Loco, n): Number of entries into the corridors up to 5 min	5.30; (1, 52) < 0.05		
2. Exploration (Expl, n): Overall number of box visits up to 5 min		8.26; (4, 52) < 0.0001	3.32; (7, 75) < 0.01
3. Expl_FR_RW: Exploration frequency into the rewarded boxes up to the collection of all rewards: (number of RW box visits) × 300/(Eat_T)	17.06; (1, 52) < 0.005		
4. **Expl_FR_NRW:** Exploration frequency into the non-rewarded boxes up to the collection of all rewards: (number of NRW box visits) × 300/(Eat_T)		3.97; (2, 26) < 0.05	2.43; (7, 75) < 0.05
5. LAT_RW: Time up to the first exploration into a rewarded box	33.25; (1, 52) < 0.0001		2.69; (7, 75) < 0.05
6. **LAT_NRW**: Time up to the first exploration into a non-rewarded box			5.28; (7, 75) < 0.0001
7. Skipping: Number of entries into the corridors before the first exploration	27.33; (1, 52) < 0.0005		3.16; (7, 75) < 0.01
8. Adequate exploration (A_E, %): (Eat_N) × 100/(number of explorations up to Eat_T)	6.59; (1, 52) < 0.05		
9. Learning capacity (L_C, %): [(Eat_N) × 300 × 100]/[(number of rewards) × (Eat_T)]	23.46; (1, 52) < 0.0005		
**10. Working memory (W_M, %):** (Eat_N) × 100/(number of exploration into the RW boxes up to collection of rewards)	4.92; (1, 26) < 0.05		
**11. Reference memory (R_M, %):** (number of exploration into the RW boxes up to collection of rewards) × 100/(number of exploration into all boxes up to collection of rewards)			3.43; (7, 75) < 0.005

**Abbreviations:** Eat_T—eating time; Eat_N—number of consumed rewards; RW—rewarded (baited); NRW—non-rewarded (non-baited). **Statistical notation:** Values are presented as F-value (degrees of freedom); *p*-value. **Note:** Parameters in bold were calculated only for trials with eight rewards (Tasks 2 and 3).

**Table 2 ijms-26-06019-t002:** Mean values from D2R binding experiments, including calculated maximal binding capacities (Bmax) and dissociation constants (Kd) in cerebral cortex and hippocampus.

Region	Treatment	B_max_ + S.E.M. (fmol/mg Protein)	K_d_ + S.E.M. (nM)
**Cortex**	LE	578.6 + 38.0	1.01 + 0.16
	Water	495.2 + 36.7	0.77 + 0.15
	CAFF	619.2 + 37.5	1.16 + 0.17
	MTF	509.0 + 41.4	0.80 + 0.17
	FURO	527.4 + 45.5	0.88 + 0.19
	CAFF_MTF	671.2 + 57.5	1.25 + 0.25
	CAFF_FURO	661.7 + 55.7	1.20 + 0.24
**Hippocampus**	LE	701.6 + 31.8	0.86 + 0.10
	Water	788.0 + 29.2	0.94 + 0.09
	CAFF	**897.5 + 26.7 **; p = 0.05 #**	1.10 + 0.08
	MTF	800.9 + 28.8	1.00 + 0.09
	FURO	**763.0 + 23.8+**	0.92 + 0.07
	CAFF_MTF	**971.5 + 30.2 ***; $; #**	1.16 + 0.09
	CAFF_FURO	**938.6 + 39.1 **; o; #**	1.06 + 0.11

Values are presented as mean ± SEM (fmol/mg protein for Bmax; nM for Kd). Bold formatting indicates significant differences compared to the following groups: Lisket-water (#), caffeine (+), metformin ($), and furosemide (o). Statistical significance is denoted as follows: #, +, $, o: **: *p* < 0.01; ***: *p* < 0.005).

## Data Availability

The data supporting the findings of this study are available from the corresponding author upon reasonable request. The data are stored in a controlled-access repository at the Department of Physiology, University of Szeged. Correspondence and requests for materials should be addressed to GH.

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
