# Peer review of "Repurposing Caffeine, Metformin, and Furosemide to Target Schizophrenia-Related Impairments in a Triple-Hit Rat Model"

_ijms, 2025, doi:10.3390/ijms26136019_

Round 1

Reviewer 1 Report

Comments and Suggestions for Authors

This research shows that non-psychotropic agents – metformin, furosemide and caffeine – seem to alleviate schizophrenia-related impairments in a triple-hi rat model. The results can be of relevance for the study of repurposing above mentioned drugs as an adjuvant therapy of schizophrenia. Of particular interest and utility is the ability of the clinically used compounds to act synergistically with each other i.e. caffeine plus metformin and caffeine and furosemide. Overall this a well-designed study with the use of the triple-hit rat model. This model was developed to closely resemble the clinical features of schizophrenia. However, some issues raised my concerns, which I have listed below:

  1. Could you explain in more detail whether the animals were in cages individually or in groups of several individuals, please compare the lines 351-352 and lines 430-432?
  2. Lines: 430-432 whether the animals were not aggressive towards each other after being grouped together after a long period of isolation especially since they were males?
  3. Were any changes in the animals' body weight observed? This is an important parameter, especially in the context of the use of metformin, which generally causes a decrease in the animals' body weight when administered over a long period of time.

Correction: line 430 “thye”.

Author Response

Responses to the Reviewer1

First of all, we would like to thank You for the insightful comments and valuable suggestions. We believe these have significantly contributed to improving the quality of our manuscript. Please find our detailed responses below. All modifications are highlighted in the “CORRECTIONS” version of the revised manuscript. We hope that the reviewers will find our revisions satisfactory.

Reviewer1

Comments and Suggestions for Authors

This research shows that non-psychotropic agents – metformin, furosemide and caffeine – seem to alleviate schizophrenia-related impairments in a triple-hit rat model. The results can be of relevance for the study of repurposing above mentioned drugs as an adjuvant therapy of schizophrenia. Of particular interest and utility is the ability of the clinically used compounds to act synergistically with each other i.e. caffeine plus metformin and caffeine and furosemide. Overall this a well-designed study with the use of the triple-hit rat model. This model was developed to closely resemble the clinical features of schizophrenia. However, some issues raised my concerns, which I have listed below:

  1. Could you explain in more detail whether the animals were in cages individually or in groups of several individuals, please compare the lines 351-352 and lines 430-432?
  2. Lines: 430-432 whether the animals were not aggressive towards each other after being grouped together after a long period of isolation especially since they were males?

Response:

  1. We apologize for the omission of this information in the original manuscript. As suggested, we have clarified the group housing conditions in the revised version. Specifically, only a brief period of fighting (approximately 10–15 minutes) was observed following resocialization of the Lisket animals, with no injuries or prolonged aggression. This has now been explicitly stated in the ‘Experimental Paradigm’ section.
  2. Thank you for this helpful suggestion. Fortunately, following resocialization, only a brief period of fighting (approximately 10–15 minutes) was observed in the Lisket animals, with no injuries or signs of prolonged aggression. This clarification has been added to the ‘Experimental Paradigm’ section of the revised manuscript (see below).

Experimental paradigm

As in our earlier studies [18,103], Lisket animals were housed individually after weaning (at 4 weeks of age) for 28 days and received intraperitoneal ketamine treatment (30 mg/kg/day for 5 days) during the second week of isolation. They were then re-housed in groups (2–3 per cage) and allowed a one-week recovery period without further interventions. Resocialization did not lead to aggressiveness or injury in these animals. Control animals were socially housed throughout and did not receive ketamine or vehicle injection, to prevent them from any stress. Baseline behavioral testing was conducted at 10 weeks of age using the Ambitus apparatus without any prior training (Ambitus1; Fig. 1B). After 48-hours of food restriction (no food with water ad libitum), two tasks were performed: Task 1 (Trials 1–2; all boxes baited; PRE_ALL phase) in the morning, and Task 2 (Trials 3–4; only internal boxes baited; PRE_IN phase) three hours later (Fig. 1B). Thus, four trials were conducted in the Ambitus 1 test on a single day. The intertrial interval was 3 hours between morning and afternoon sessions and 2 minutes between trials within each session. At least one week later, at the age of three months, all rats were single housed during the drug administration period to allow accurate monitoring of individual fluid consumption. One control group (LE_W; water only) and eight Lisket groups (based on drug treatment) were included in a 14-day protocol (days 1-14). Lisket groups received water (Lisket_W), caffeine (CAFF, 0.5 mg/ml), metformin (MTF, 1 mg/ml), furosemide (FURO, 0.1 mg/ml), or combinations (CAFF_MTF, CAFF_FURO, MTF_FURO, and TRIPLE; Fig. 1B). Drug doses were based on previous rodent studies that showed effectiveness and were approximately equivalent to human therapeutic levels [30,31,104,105]. Animals were assigned to groups using stratified randomization based on baseline behavioral scores and body weight to ensure comparability. A second Ambitus test (Ambitus 2) was conducted on days 11-14, following two days of food deprivation (no food, water ad libitum on days 9–10), with four trials per day. Testing phases were defined by the task used: POST_ALL (Task 1, Trials 1–2, day 11), POST_EX (Task 3, Trials 3–4 on day 11 and Trials 5–8 on day 12), and POST_IN (Task 2, Trials 9–16 on days 13–14). In total, 16 post-treatment trials were conducted over four days, using the same intertrial interval as in Ambitus1. Moderate food restriction was maintained throughout testing (about 25 mg/kg/day), and drinking fluid was available ad libitum. Body weight was recorded weekly, and total fluid (and drug) intake, normalized to body weight, was calculated. Behavioral experiments were conducted between 8:00 AM and 4:00 PM under dim lighting.

On day 15, animals were decapitated, and brains were rapidly removed. The CTX and hippocampus were immediately dissected, frozen in liquid nitrogen, and stored at –75 °C until used in in vitro assays. Based on behavioral results, the MTF_FURO and TRIPLE groups were excluded from in vitro analysis.“

  1. Were any changes in the animals' body weight observed? This is an important parameter, especially in the context of the use of metformin, which generally causes a decrease in the animals' body weight when administered over a long period of time.

Response:

Thank you for raising this point. As stated in the Results section, no significant differences in body weight were observed between the various drug-treated Lisket groups. However, we did identify and correct a transcription error in the F-value. The corrected sentence now reads:

“The body weight of the Lisket animals was significantly lower throughout the entire experiment (Group: F(1,174) = 14.22; P < 0.001), with no significant differences observed between the various drug-treated Lisket groups during the treatment (F(7,150) = 0.26; P = 0.97).’”

Correction: line 430 “thye”.

Response:

Thank you for pointing out this typographical error. The misspelling (‘thye’) has been corrected in the revised manuscript. 

Reviewer 2 Report

Comments and Suggestions for Authors

Revision

Abstract

-Clarify treatment vehicle: Please specify the vehicle used for drug administration (e.g., water, saline, etc.), as this is essential for reproducibility and interpretation.

-Revise long sentence for clarity: This study evaluated their behavioral effects individually and in combination using the reward-based Ambitus test, and assessed cerebral D2 dopamine receptor (D2R) expression and binding in a schizophrenia-like triple-hit animal model (Lisket rats), derived from the Long Evans (LE) strain.” is too long and potentially confusing. Consider splitting it for clarity.

-Use of the term “non-psychotropic drugs”: The term is inaccurate in this context, as caffeine is psychoactive by definition—it stimulates the central nervous system, enhances alertness, and affects mood. A more appropriate alternative could be: “non-antipsychotic agents” or “drugs not originally developed for psychiatric disorders.”

-Clarify this sentence: “Metformin and furosemide reduced exploratory behavior but improved reference memory; these effects were moderated by caffeine co-administration.” While this aligns with the results (e.g., reduced visits but improved correct entries), the term “moderated” might be slightly ambiguous, as caffeine co-administration in some cases reversed, and in others nullified, the effects.

-Regarding this sentence: “The metformin–furosemide and triple combinations had no behavioral impact versus water-treated Lisket rats.”This is technically accurate based on the statistical outcomes. However, since the discussion highlights behavioral trends (e.g., the triple combination tending to reduce visits), you may consider acknowledging those tendencies in the phrasing: “Although no statistically significant behavioral differences were found compared to water-treated Lisket rats, some trends were observed, such as a reduction in exploratory visits in the triple combination group.”

Introduction

-Caffeine as an adjuvant in schizophrenia

The statement “Caffeine may alleviate negative symptoms of schizophrenia” requires a more nuanced formulation. While some studies have proposed potential benefits, the overall evidence remains controversial. Caffeine has also been reported to exacerbate anxiety, psychosis, and may interfere with antipsychotic efficacy.  Consider revising this sentence to: “Caffeine has been proposed to influence negative symptoms of schizophrenia” or “may modulate negative symptoms”. Avoid asserting unqualified beneficial effects without acknowledging conflicting evidence.

-Diuretics and schizophrenia

The phrase “including... schizophrenia” when listing the beneficial effects of diuretics may be misleading. The evidence supporting the use of loop diuretics (such as bumetanide) in schizophrenia is limited and primarily indirect, often related to GABAergic modulation, which is not equivalent to furosemide’s mechanisms. A more accurate formulation might be: “...have been explored in CNS disorders such as epilepsy, anxiety, autism spectrum disorders, and, to a lesser extent, schizophrenia.”

-Validation of the Lisket model

The description of the Lisket model is comprehensive. However, caution should be exercised with assertive language regarding its translational value. For example, the statement “The Lisket model was designed to retain strong face validity for schizophrenia” may overstate the current evidence. A more rigorous alternative could be: “...was designed to enhance face validity for schizophrenia-like behaviors.” Unless extensive comparative validation studies are available, it is advisable to avoid categorical claims.

-Dopaminergic signaling and D2R in schizophrenia

The emphasis on D2 receptor alterations in the cortex and hippocampus is relevant. However, interpretations should be made carefully. Schizophrenia is typically associated with mesolimbic hyperdopaminergia and cortical hypodopaminergia. Please ensure that changes in D2R binding are discussed with appropriate nuance (e.g., does increased binding reflect higher receptor availability, reduced endogenous dopamine levels, or both?). This distinction is important for contextualizing the findings.

Results

-2.1 General observations

Statistical inconsistency: The reported statistic “F(7,150) = 10.26; P = 0.97” is not compatible. An F-value of 10.26 with df = (7,150) would yield P ≪ 0.01. Verify whether the F-value or the P-value was mis–transcribed. It is most likely that P < 0.001.

Drug doses calculated from consumption: You note that the CAFF_FURO group received a higher dose than the monotherapy groups. This presumably reflects greater total fluid intake, but the cause (e.g., increased palatability, diuretic-driven thirst, pharmacological interactions) is unclear. Add a clarifying sentence such as: “The higher drug dose in the CAFF_FURO group resulted directly from increased fluid consumption, likely due to enhanced palatability or compensatory drinking following diuresis.”

“No significant effects on metformin dose”: It is ambiguous what “effects” refers to here. Do you mean that metformin consumption (mg/kg/day) did not differ between treatment groups? Rephrase for clarity.

Clarify group comparison for mean fluid intake: The phrase “compared to the other groups (mean: 104 ± 3.0 mL/kg/day)” could be misread as including CAFF_FURO and TRIPLE.  Specify which groups are included.

-2.2 Behavioral differences between water-drinking LE and Lisket animals

Table 1 formatting: The table is difficult to interpret—unclear whether the values are F-statistics, z-scores, or means ± SEM. Label each column explicitly (e.g., “F(df)”, “P-value”, “Mean ± SEM”) and ensure consistency with the text.

Phase effects vs. interaction (z-scores): You state, “None of the z-scores showed significant effects for phase alone,” yet describe phase-dependent changes in exploration. Clarify that changes across phases arise from a group × phase interaction, not a main effect of phase.

Latency and reference memory exception: The sentence “Except for the latency of the first non-baited box visit (Lat_NRW) and reference memory (R_M).” is incorrect with respect to Lat_NRW. In Figure 2, Lat_NRW does show significant group differences in phases 4 and 5. Remove Lat_NRW from this exception.

Locomotion and working memory timing: The statement “Group differences in locomotion and working memory parameters were observed only during the final phase” contradicts Figure 2, where LOCO and Expl_FR_RW differ from phase 1 onward. Revise to: “Group differences in locomotion and working memory parameters were present from the early phases and became more pronounced by the final phase.”

-2.3 Behavioral results of pharmacological treatments in Lisket animals

ANOVA description:  Instead of “One-way ANOVA revealed significant treatment effects for several parameters (2, 4, 5, 6, 7, and 11)…” Specify the behaviors directly.

Reference memory baseline vs. treatment effects: You note that R_M shows no LE vs. Lisket difference but then report that MTF and FURO improve R_M. Confirm whether Lisket_W and LE_W baselines were statistically equivalent. If so, state: “Although baseline R_M did not differ between Lisket_W and LE_W, both metformin and furosemide treatments produced significant improvements relative to Lisket_W.”

Skipping behavior as an attention proxy: You interpret increased skipping as attentional impairment, but this requires justification. Include in Methods or Discussion a brief rationale or citation for using skipping frequency as an index of attentional deficit.

Specify “working memory parameters”: The phrase “working memory parameters” is vague.

Ambiguity in combination effects: The description “attenuated effects compared to the individual drugs, with most values resembling those of the water-drinking Lisket group” leaves unclear whether the combination nullified benefits or was entirely inert.

Caffeine monotherapy – exploration vs. Lat_NRW: You state “Caffeine monotherapy caused a moderate increase in exploratory behavior, accompanied by a significantly shorter Lat_NRW,” but Figure 3A shows no significant increase in exploration.

Furosemide effects on Lat_NRW: The sentence “Only furosemide significantly increased the Lat_NRW” is incorrect: furosemide decreased Lat_NRW.

Figure 4B and supplementary material: You should include a representative gel image in Fig. 4B, and upload the full original gels as supplementary files.

Discussion

-Avoid unnecessary repetitions: There are several repeated concepts already mentioned in the Introduction or Results sections. For example: "...the hippocampus and the cortex, which operate through several neurotransmitter systems such as dopamine, glutamate, GABA, and adenosine..."

-Elaborate on the functional implications of D2R: "...the exact implications of the enhanced cortical D2R protein expression remain unclear..." Consider expanding on what this upregulation might mean functionally.

-Pharmacological combinations: more explicit development. The section on drug combinations is interesting but somewhat brief. For instance: "...the behavioral outcomes of the MTF_FURO and TRIPLE combinations resembled those of the water-drinking group..."  Briefly discuss the possible implications of this observation. Does it suggest loss of efficacy? Interference between mechanisms?

-Clarify the type of learning affected: "While none of the treatments had a significant effect on learning capacity" It would be helpful to specify whether this refers to acquisition learning or working memory, depending on the test used.

-Regarding caffeine: "Caffeine treatment alone enhanced activity-related parameters and increased D2R binding, and was associated with a tendency toward decreased reference memory..." It is appropriate to report a "tendency" only if the data do not reach statistical significance. If the difference is statistically significant, it should be clearly stated.

-Regarding metformin and furosemide: "Metformin and furosemide reduced exploratory activity and attentional performance, while improving reference memory..." This sentence combines multiple effects. Clarify if both drugs reduce exploration and attention, or if one affects one parameter and the other another. If attentional changes were not clearly observed or directly assessed, this point should be moderated.

-Drug dose equivalence to humans: Suggest adding the human-rat dose conversion or scaling information to contextualize the relevance of the doses used.

-Line 278: "However, no change in receptor protein expression was observed in this region." Specify if "this region" refers to the hippocampus. Also, if increased binding is observed without changes in protein expression, it would be useful to propose possible explanations (e.g., changes in receptor affinity, receptor reserve) as is done later for furosemide.

-Line 298: "...no significant D2R-related changes werre observed in Lisket rats [with metformin]" Comment: Correct the typo ("werre"). If there is any subtle effect or trend, it could be briefly mentioned.

-Line 305: "Furosemide, like metformin, improved reference memory, but reduced the animals’ behavioral activity." This is appropriate if the reduction in activity was statistically significant; otherwise, consider softening this to "a tendency to reduce..."

-Lines 331–332: "The CAFF_FURO combination produced similar changes in receptor expression but was associated with different behavioral outcomes." This is an interesting dissociation; it would be helpful to propose tentative hypotheses explaining why molecular and behavioral changes diverge.

-Lines 333–343 (combinations): Good point about antagonistic interactions in combinations, but emphasize that this complicates mechanistic interpretation (as mentioned). Brief speculation on possible pharmacokinetic effects (e.g., altered bioavailability due to diuresis) or differences in consumption between groups would be valuable.

-General suggestions:

Reinforce that Bmax and protein expression (Western blot) do not always correspond directly, as they can be influenced by changes in receptor affinity, internalization, or trafficking.

Consider adding a concluding sentence highlighting the significance of these findings for future research and clinical treatment strategies, especially regarding common drug combinations in patients.

Methods

-Overall structure: 

The Experimental Paradigm (4.4) should appear immediately after the Animals (4.1) section. This would improve the logical flow of the manuscript and help the reader understand the experimental design earlier.

Please revise to eliminate redundant information (e.g., details about sacrifice appear in multiple places).

Suggest restructuring the entire section to follow a logical progression: Animals → Experimental Paradigm → Behavioral Testing → Sacrifice and Biochemical Analyses → Statistical Analyses.

Figure 1B could be clarified. Green arrows: The two green arrows indicating "Ambitus 1" and "Ambitus 2" should be the same size. The current difference in size might misleadingly suggest that one test is more important or extended than the other, when both represent equivalent pre- and post-treatment evaluations. Empty orange rectangle: There is an empty orange box next to the one labeled "Ambitus test 1" — could you clarify its purpose or remove it if it’s not relevant? As it stands, it creates visual confusion. Color-coded task labels: It would be helpful to indicate in the figure legend what each color represents in the lower section (e.g., green for Task 1, pink for Task 2, blue for Task 3), to improve clarity for the reader.

-4.1 Animals

Sex, age, and group size: Please specify the sex and age of the animals. Also, include the total number of animals used and the number allocated to each group.

Terminological consistency: LE rats are referred to as “control,” but it would be clearer to explicitly state that they were not subjected to the experimental model (i.e., no isolation or ketamine).

-4.2 Behavioral testing

Behavioral procedures: Please add that the apparatus was cleaned between animals (e.g., with 70% ethanol or acetic acid) to avoid olfactory cues. Also specify whether animals were habituated to the behavioral room, and for how long prior to testing. Video analysis was performed by at least two researchers who were blinded to the animals' treatment conditions? Add this information to manuscript.

Number of trials per animal: It is essential to state how many sessions each animal underwent and whether there was a habituation or pretraining period. This is particularly important if learning or memory parameters are being analyzed.

Clarity of apparatus description: You mention “four side boxes per corridor” (two inner and two outer), with a total of 16 boxes. This implies there are four corridors, but it is not clearly stated.

Food motivation: Since the Ambitus test is reward-based, clarify whether animals were food-restricted prior to testing. This is crucial for interpreting behavioral outcomes.

Manufacturer website citation: Please ensure the manufacturer’s website is cited following the journal’s formatting guidelines.

-4.3 Biochemical methods 4.3.1 Membrane preparations

What method of sacrifice was used? Please specify.

What was the sample size (n) used for neuronal membrane preparations in each group?

-4.3.2 Radioligand binding

Sentence: “In saturation binding experiments we use radiolabeled ligands in increasing concentrations and measure the amount of specifically bound radioactive ligand in the function of the applied radioligand concentrations.” This sentence is too long and awkward. Also, “in the function of” is not appropriate in scientific English. Suggested revision: “In saturation binding experiments, radiolabeled ligands were applied at increasing concentrations, and the amount of specifically bound radioactivity was measured as a function of ligand concentration.”

Terminology: “5-HT2 serotoninergic receptors” → Use the more widely accepted term: “5-HT2 serotonergic receptors.”

Clarify filtration procedure: “…through Whatman GF/B glass fibers” → Consider changing to:

“…through Whatman GF/B glass fiber filters” for clarity.

-4.3.3 Western blot

Please unify centrifugation units throughout the manuscript (use either ×g or rpm consistently). E.g., “11.000×g” → use a comma, not a point (11,000×g).

Missing information: Electrophoresis and transfer conditions: temperature and duration.

Blocking step: composition of the blocking buffer, incubation time and temperature.

Antibody incubation: Were primary antibodies incubated simultaneously? Which secondary antibodies were used? Incubation times and temperatures?

Was the optical density normalized to β-actin?

-4.4 Experimental paradigm

Group assignment: You state that animals were assigned based on baseline behavioral scores and body weight. Consider specifying whether this was stratified randomization or block matching.

Control design: Ideally, control animals should have undergone the same isolation and vehicle injection protocol as the experimental groups to control for handling and social stress. Please justify why this was not done, considering the gregarious nature of rodents.

Food restriction: Provide the amount of food given per animal per day, expressed in g/kg/day.

Drug administration: It is essential to state clearly in this section that drugs were administered via drinking water. This detail only appears in the abstract. Also, provide the source/manufacturer of each drug.

Testing conditions, add this information to manuscript:

What time of day were the trials conducted?

How long did each trial last?

What was the inter-trial interval?

-4.5 Measurements and statistical analyses

Repeated measures ANOVA: Please indicate whether the sphericity assumption was tested. If violated, which correction method was applied (e.g., Greenhouse-Geisser)? Add this information to manuscript.

Clarify statistical phrasing: “Calculated separately for the five phases (PRE_ALL, PRE_IN, POST_ALL, POST_EX, and POST_IN; Fig. 1B).”  Better phrasing: “Calculated separately for each of the five phases…”

“Significance levels of the behavioral parameters analyzed”.  “Significance levels” refers to statistical thresholds (e.g., p < 0.05). Use “significant differences” or “results” instead.

Trial configuration: “Parameters could only be calculated for trials in which rewards were placed on one side.” Specify whether you refer to an asymmetric reward configuration or lateralized trials.

Averaging z-scores: “Mean z-score across the POST_EX and POST_IN phases” Justify the rationale for averaging across these two distinct phases. This may be appropriate, but requires conceptual explanation.

Binding analysis: When mentioning “non-linear regression” and “one-site specific binding”, it may help to briefly state that this corresponds to the classic hyperbolic saturation model assuming a single binding site.

“Based on total protein content, radioligand concentration, and specific activity” → You may consider including the actual formula used to calculate Bmax (fmol/mg), unless already described elsewhere.

Let me know if you'd like me to prepare a condensed version for submission or convert it into a track-changes comment list.

Conclusion

-"Comprehensive behavioral and D2R characterization": This might sound overly ambitious if the D2R assessment was limited (e.g., only Western blot and no functional analysis). You could tone it down if needed, for example: “…a detailed behavioral and receptor-level assessment…”

-"Moderate D2R alterations": Consider specifying whether these were increases or decreases, and in which brain regions, to provide greater clarity and impact.

-"Repurposing of common drugs": It may be more precise to indicate the drug classes involved (e.g., anxiolytics, neuromodulators, etc.) to avoid ambiguity.

Graphical Abstract

-Furosemide increases D2R levels in the cortex, but the graphic only depicts 'hippocampal D2R binding'. This is not an error, but rather a simplification that omits part of the result. For full accuracy, a separate column for cortical D2R could be added, or an asterisk could be used to indicate that cortical effects were also observed.

Comments on the Quality of English Language

The manuscript addresses a relevant topic; however, I strongly recommend a thorough revision of the English writing.

Author Response

Responses to the Reviewer2

First of all, we would like to thank You for the insightful comments and valuable suggestions. We believe these have significantly contributed to improving the quality of our manuscript. Please find our detailed responses below. All modifications are highlighted in the “CORRECTIONS” version of the revised manuscript. We hope that the reviewers will find our revisions satisfactory.

Abstract

-Clarify treatment vehicle: Please specify the vehicle used for drug administration (e.g., water, saline, etc.), as this is essential for reproducibility and interpretation.

Response: Thank you for this suggestion. Due to word count limitations in the Abstract, we clarified the treatment vehicle in the ‘Animals’ section of the Methods. As stated there, all drugs were dissolved in tap water and administered via the drinking bottle (see below). The revised section reads:

“A larger number of animals was used in the drug-treated groups to account for expected variability due to individual differences in drug sensitivity. All drugs were dissolved in tap water and administered via the drinking bottle.”

-Revise long sentence for clarity: This study evaluated their behavioral effects individually and in combination using the reward-based Ambitus test and assessed cerebral D2 dopamine receptor (D2R) expression and binding in a schizophrenia-like triple-hit animal model (Lisket rats), derived from the Long Evans (LE) strain.” is too long and potentially confusing. Consider splitting it for clarity.

Response: Thank you for the helpful suggestion. We revised the sentence in the Abstract by splitting it into shorter, clearer statements. The revised version now reads:

“This study evaluated the effects of these agents on behavioral parameters using the reward-based Ambitus test, and on the cerebral D2 dopamine receptor (D2R) expression and binding. The drugs were administered individually and in combination in a schizophrenia-like triple-hit animal model (Lisket rats), derived from the Long Evans (LE) strain.”

-Use of the term “non-psychotropic drugs”: The term is inaccurate in this context, as caffeine is psychoactive by definition—it stimulates the central nervous system, enhances alertness, and affects mood. A more appropriate alternative could be: “non-antipsychotic agents” or “drugs not originally developed for psychiatric disorders.”

Response: We agree with the reviewer’s observation and have replaced ‘non-psychotropic drugs’ with the more accurate term ‘non-antipsychotic agents’ in the Abstract.

-Clarify this sentence: “Metformin and furosemide reduced exploratory behavior but improved reference memory; these effects were moderated by caffeine co-administration.” While this aligns with the results (e.g., reduced visits but improved correct entries), the term “moderated” might be slightly ambiguous, as caffeine co-administration in some cases reversed, and in others nullified, the effects.

Response: Thank you for this helpful comment. To improve clarity, we replaced the term ‘moderated’ with ‘inhibited’ to better reflect the reversing or nullifying effects of caffeine co-administration.

-Regarding this sentence: “The metformin–furosemide and triple combinations had no behavioral impact versus water-treated Lisket rats.”This is technically accurate based on the statistical outcomes. However, since the discussion highlights behavioral trends (e.g., the triple combination tending to reduce visits), you may consider acknowledging those tendencies in the phrasing: “Although no statistically significant behavioral differences were found compared to water-treated Lisket rats, some trends were observed, such as a reduction in exploratory visits in the triple combination group.”

Response: Thank you for the suggestion. We revised the sentence in the Abstract to acknowledge the observed behavioral trend, while maintaining the correct statistical interpretation. The corrected sentence now reads:

‘Although no statistically significant behavioral differences were found compared to water-treated Lisket rats, a trend toward reduced exploratory visits was observed in the triple-combination group.’

Introduction

-Caffeine as an adjuvant in schizophrenia

The statement “Caffeine may alleviate negative symptoms of schizophrenia” requires a more nuanced formulation. While some studies have proposed potential benefits, the overall evidence remains controversial. Caffeine has also been reported to exacerbate anxiety, psychosis, and may interfere with antipsychotic efficacy.  Consider revising this sentence to: “Caffeine has been proposed to influence negative symptoms of schizophrenia” or “may modulate negative symptoms”. Avoid asserting unqualified beneficial effects without acknowledging conflicting evidence.

Response: Thank you for the suggestion. We have revised the statement to use more balanced language, replacing ‘may alleviate’ with ‘may modulate,’ to reflect the mixed evidence in the literature. The revised sentence now reads:

‘Caffeine may also modulate negative symptoms of schizophrenia and reduce the extrapyramidal and sedative side effects of antipsychotics.’

-Diuretics and schizophrenia

The phrase “including... schizophrenia” when listing the beneficial effects of diuretics may be misleading. The evidence supporting the use of loop diuretics (such as bumetanide) in schizophrenia is limited and primarily indirect, often related to GABAergic modulation, which is not equivalent to furosemide’s mechanisms. A more accurate formulation might be: “...have been explored in CNS disorders such as epilepsy, anxiety, autism spectrum disorders, and, to a lesser extent, schizophrenia.”

Response: Thank you for the clarification. We have revised the sentence as suggested to reflect the more limited and indirect evidence for diuretics in schizophrenia. The updated version reads:

‘Diuretics have demonstrated beneficial effects in a range of CNS disorders, including epilepsy, anxiety disorders, autism spectrum disorder, and, to a lesser extent, schizophrenia.’

-Validation of the Lisket model

The description of the Lisket model is comprehensive. However, caution should be exercised with assertive language regarding its translational value. For example, the statement “The Lisket model was designed to retain strong face validity for schizophrenia” may overstate the current evidence. A more rigorous alternative could be: “...was designed to enhance face validity for schizophrenia-like behaviors.” Unless extensive comparative validation studies are available, it is advisable to avoid categorical claims.

Response: Thank you for this important suggestion. We have adjusted the wording to avoid overstating the model’s validity. The revised sentence now reads:

‘The Lisket model was designed to enhance face validity for schizophrenia-like behaviors, while avoiding the marked inactivity observed in Wistar and Wisket animals.’

-Dopaminergic signaling and D2R in schizophrenia

The emphasis on D2 receptor alterations in the cortex and hippocampus is relevant. However, interpretations should be made carefully. Schizophrenia is typically associated with mesolimbic hyperdopaminergia and cortical hypodopaminergia. Please ensure that changes in D2R binding are discussed with appropriate nuance (e.g., does increased binding reflect higher receptor availability, reduced endogenous dopamine levels, or both?). This distinction is important for contextualizing the findings.

Response: Thank you for this suggestion. To address the need for greater nuance, we added the following sentence to the Introduction:

‘Dopamine dysfunction in schizophrenia affects several brain regions, notably the cerebral cortex (CTX) and hippocampus, and includes decreased dopamine levels in these areas accompanied by increased D2 receptor density.’

-2.1 General observations

Statistical inconsistency: The reported statistic “F(7,150) = 10.26; P = 0.97” is not compatible. An F-value of 10.26 with df = (7,150) would yield P ≪ 0.01. Verify whether the F-value or the P-value was mis–transcribed. It is most likely that P < 0.001.

Response: Thank you for identifying this inconsistency. This was indeed a transcription error. The correct F-value is 0.26, and we have revised the sentence accordingly in the Results section:

‘The body weight of the Lisket animals was significantly lower throughout the entire experiment (Group: F(1,174) = 14.22; P < 0.001), with no significant differences observed between the various drug-treated Lisket groups during the treatment (F(7,150) = 0.26; P = 0.97).’

Drug doses calculated from consumption: You note that the CAFF_FURO group received a higher dose than the monotherapy groups. This presumably reflects greater total fluid intake, but the cause (e.g., increased palatability, diuretic-driven thirst, pharmacological interactions) is unclear. Add a clarifying sentence such as: “The higher drug dose in the CAFF_FURO group resulted directly from increased fluid consumption, likely due to enhanced palatability or compensatory drinking following diuresis.”

Response: Thank you for the suggestion. We have clarified the reason for the elevated drug doses in the CAFF_FURO group. As there are no data on the palatability of this combination, we attributed the increased intake to compensatory drinking due to diuresis. The revised sentence reads:

‘The higher drug dose in this group likely resulted from increased fluid consumption, presumably due to compensatory drinking following diuresis.’

“No significant effects on metformin dose”: It is ambiguous what “effects” refers to here. Do you mean that metformin consumption (mg/kg/day) did not differ between treatment groups? Rephrase for clarity.

Response: Thank you for pointing this out. We agree that the sentence was unclear and, upon review, determined that it added no meaningful information. Since none of the monotherapies—including metformin—significantly altered fluid or drug intake compared to the water-drinking group, we have removed the sentence from the manuscript.

Clarify group comparison for mean fluid intake: The phrase “compared to the other groups (mean: 104 ± 3.0 mL/kg/day)” could be misread as including CAFF_FURO and TRIPLE.  Specify which groups are included.

Response: Thank you for noting this potential ambiguity. We clarified the sentence to specify which groups were included in the average fluid intake comparison. The revised version now reads:

‘Post hoc analysis revealed that seven groups (CAFF, MTF, FURO, CAFF_MTF, MTF_FURO, TRIPLE) did not differ significantly from one another, consuming a similar daily volume (mean: 104 ± 3.0 mL/kg/day). In contrast, the CAFF_FURO-treated group (146 ± 11.5 mL/kg/day) drank significantly more fluid than the other groups, except for the TRIPLE-treated group (126 ± 13.7 mL/kg/day).’

2.2 Behavioral differences between water-drinking LE and Lisket animals

Table 1 formatting: The table is difficult to interpret—unclear whether the values are F-statistics, z-scores, or means ± SEM. Label each column explicitly (e.g., “F(df)”, “P-value”, “Mean ± SEM”) and ensure consistency with the text.

Response: Thank you for highlighting this formatting issue. We revised the title and structure of Table 1 to clarify the meaning of each column and ensure alignment with the text. The updated title now reads:

‘Table 1. Definitions and calculation methods (first column) along with statistical results for behavioral parameters, analyzed by group (second and third columns; LE vs. Lisket; repeated-measures ANOVA) and by pharmacological treatment in Lisket animals (fourth column; one-way ANOVA).’

Phase effects vs. interaction (z-scores): You state, “None of the z-scores showed significant effects for phase alone,” yet describe phase-dependent changes in exploration. Clarify that changes across phases arise from a group × phase interaction, not a main effect of phase.

Response: Thank you for this insightful comment. We clarified the distinction between phase effects and group × phase interactions in the Results section. The revised text now reads:

‘Repeated-measures ANOVA revealed significant effects of group or group-by-phase interaction for most parameters, except for the latency of the first non-baited box visit (Lat_NRW) and reference memory (R_M). However, none of the z-scores showed significant effects for phase alone (Table 1).’

Latency and reference memory exception: The sentence “Except for the latency of the first non-baited box visit (Lat_NRW) and reference memory (R_M).” is incorrect with respect to Lat_NRW. In Figure 2, Lat_NRW does show significant group differences in phases 4 and 5. Remove Lat_NRW from this exception.

Locomotion and working memory timing: The statement “Group differences in locomotion and working memory parameters were observed only during the final phase” contradicts Figure 2, where LOCO and Expl_FR_RW differ from phase 1 onward. Revise to: “Group differences in locomotion and working memory parameters were present from the early phases and became more pronounced by the final phase.”

Response: Thank you for pointing out these inconsistencies. Upon review, we discovered that Figure 2 previously omitted key parameters, including Expl_FR_NRW and WM, which caused a mismatch between the figure and the text. We have corrected Figure 2 and its legend accordingly. Additionally, we revised the text to accurately reflect the timing and presence of group differences in locomotion and working memory. Notably, Lat_NRW was not included in Figure 2, as it did not show a significant group or group-by-phase interaction effect in the final analysis. See the corrected figure below.

Figure 2. Behavioral parameters showing significant effects of group or group by phase interaction (expressed as Z-scores) between LE-water and Lisket-water animals across the five phases of the Ambitus test. Definitions, abbreviations, calculation methods, and statistical results for these parameters are provided in Table 1. Asterisks (*) indicate significant differences between groups (P < 0.05). Note that values indicating enhanced behavioral activity or learning functions have positive Z-scores, whereas parameters where higher raw values reflect impairments (e.g. latency, skipping) are shown with negative Z-scores (e.g. a high latency value corresponds to a negative Z-score).

-2.3 Behavioral results of pharmacological treatments in Lisket animals

ANOVA description:  Instead of “One-way ANOVA revealed significant treatment effects for several parameters (2, 4, 5, 6, 7, and 11)…” Specify the behaviors directly.

Response: Thank you for this helpful suggestion. We have revised the text to refer to the specific behavioral parameters by name rather than number. The updated sentence now reads:

‘One-way ANOVA revealed significant treatment effects for several parameters (Expl, Expl_FR_NRW, Lat_RW, Lat_NRW, skipping, and R_M), as shown in Table 1 and Figure 3.’

Reference memory baseline vs. treatment effects: You note that R_M shows no LE vs. Lisket difference but then report that MTF and FURO improve R_M. Confirm whether Lisket_W and LE_W baselines were statistically equivalent. If so, state: “Although baseline R_M did not differ between Lisket_W and LE_W, both metformin and furosemide treatments produced significant improvements relative to Lisket_W.”

Response: Thank you for this observation. We reanalyzed the data and confirmed that there was no significant difference in reference memory (R_M) between LE_W and Lisket_W groups. Accordingly, we revised the sentence as follows:

‘Although baseline R_M did not differ between Lisket_W and LE_W, both metformin and furosemide monotherapies significantly enhanced reference memory values compared to the water- and caffeine-treated groups. These improvements were accompanied by a moderate reduction in overall exploration.’

Skipping behavior as an attention proxy: You interpret increased skipping as attentional impairment, but this requires justification. Include in Methods or Discussion a brief rationale or citation for using skipping frequency as an index of attentional deficit.

Response: Thank you for this important point. We added a brief explanation to the Methods section (Ambitus test) to clarify the use of ‘skipping’ behavior as an index of attention. The added sentence reads:

‘As the Ambitus is a custom-designed, square-shaped corridor with side-boxes, it can detect attentional deficits by measuring skipping—defined as the number of corridors walked without exploration.’

Specify “working memory parameters”: The phrase “working memory parameters” is vague.

Response: Thank you for the suggestion. To clarify the use of the term ‘working memory parameters,’ we revised the Methods section (Ambitus test) to specify that working memory (W_M) reflects short-term memory, while reference memory (R_M) reflects long-term memory. These definitions are now included in the sentence:

‘…working memory (W_M, reflecting short-term memory), and reference memory (R_M, reflecting long-term memory)...’

Ambiguity in combination effects: The description “attenuated effects compared to the individual drugs, with most values resembling those of the water-drinking Lisket group” leaves unclear whether the combination nullified benefits or was entirely inert.

Response: Thank you for pointing out this ambiguity. We revised the sentence in the Results section to better reflect the observed lack of additive effects. The updated version reads:

‘Surprisingly, the MTF_FURO and TRIPLE combinations produced attenuated effects compared to the individual drugs, with most values resembling those of the water-drinking Lisket group—suggesting some form of antagonism between the ligands.’

Caffeine monotherapy – exploration vs. Lat_NRW: You state “Caffeine monotherapy caused a moderate increase in exploratory behavior, accompanied by a significantly shorter Lat_NRW,” but Figure 3A shows no significant increase in exploration.

Response: Thank you for pointing this out. We clarified the sentence to reflect that the increase in exploratory behavior with caffeine was not statistically significant, while the effect on Lat_NRW was. The revised sentence reads:

‘Caffeine monotherapy caused a moderate, but non-significant increase in exploratory behavior, accompanied by a significantly shorter latency to the first visit to a non-baited box (Lat_NRW) compared to the water-drinking Lisket group (Lisket_W).’

Furosemide effects on Lat_NRW: The sentence “Only furosemide significantly increased the Lat_NRW” is incorrect: furosemide decreased Lat_NRW.

Response: Thank you for pointing this out. The apparent discrepancy is due to the directionality of z-score transformations. For parameters where higher raw values indicate impairment (e.g., latency, skipping), z-scores were inverted so that lower (i.e., improved) latencies yield positive z-scores. This was not initially described clearly.

To address this, we added the following explanation to the figure legends and Methods section:

‘Values reflecting enhanced behavioral activity or learning functions were assigned positive Z-scores, while variables where higher raw values indicated impairments (e.g., latency, skipping) were inverted, resulting in negative z-scores (e.g., a high latency value was expressed as a negative z-score).’

Figure 4B and supplementary material: You should include a representative gel image in Fig. 4B, and upload the full original gels as supplementary files.

Response: Thank you for the suggestion. We have included a representative Western blot image in Figure 4B, and the full, uncropped original gels have been prepared and will be submitted as supplementary material.

Discussion

-Avoid unnecessary repetitions: There are several repeated concepts already mentioned in the Introduction or Results sections. For example: "...the hippocampus and the cortex, which operate through several neurotransmitter systems such as dopamine, glutamate, GABA, and adenosine..."

Response:Thank you for the observation. We revised the sentence in the Discussion to avoid repetition of information already presented in the Introduction. The new version reads:

‘Multiple brain regions (including the hippocampus and CTX), which are involved in cognitive functions and operate through several neurotransmitters, interact to form the complex pathophysiology of schizophrenia.’

Elaborate on the functional implications of D2R: "...the exact implications of the enhanced cortical D2R protein expression remain unclear..." Consider expanding on what this upregulation might mean functionally.

Response: Thank you for the suggestion. We expanded the discussion to consider the potential functional implications of increased cortical D2R expression. The revised text now reads:

‘The exact implications of the enhanced cortical D2R protein expression following furosemide treatment remain unclear, particularly since the CAFF_FURO combination produced similar expression changes but divergent behavioral outcomes. It is possible that the functionality of D2R is differentially modulated by these ligands. Further in vitro studies are required to explore the specific role of furosemide in D2R regulation and signaling.’

-Pharmacological combinations: more explicit development. The section on drug combinations is interesting but somewhat brief. For instance: "...the behavioral outcomes of the MTF_FURO and TRIPLE combinations resembled those of the water-drinking group..."  Briefly discuss the possible implications of this observation. Does it suggest loss of efficacy? Interference between mechanisms?

Response:

Thank you for highlighting this important point. We expanded the Discussion to reflect the possibility that the MTF_FURO and TRIPLE combinations produced antagonistic or nullifying effects, as their behavioral profiles closely resembled those of the water-treated group. We now discuss both pharmacodynamic interactions (e.g., opposing actions on neurotransmitter systems) and pharmacokinetic factors (e.g., altered bioavailability due to diuresis) as potential contributors to this lack of efficacy. See modified section below.

‘Regarding the drug combinations, caffeine co-administered with metformin or furosemide reversed the effects of these compounds on exploratory activity and reference memory. These antagonistic behavioral interactions were accompanied by increased D2R binding in both brain regions without changes in receptor protein expression. A possible explanation for the discrepancies between the binding and D2R protein expression may that enhanced receptor expression is accompanied by decreased ligand binding affinity, or that altered binding occurs despite normal protein expression. Furthermore, the binding capacity values in radioligand binding assays do not always parallel those in protein expression studies performed by Western blot analysis, as they can be influenced by various, partly experimental factors, such as receptor affinity, receptor occupancy, ionic environment and regulation, presence or absence of guanine nucleotides, allosteric interactions, internalization, trafficking, as well. These results suggest that D2R affinity for its ligand may be enhanced by these treatments, depending at least on part on the dopamine levels. The elevated exploratory activity in the CAFF_FURO group may reflect higher drug intake due to increased fluid consumption, likely compensating for the enhanced diuretic effect [101]. Surprisingly, the behavioral outcomes of the MTF_FURO and TRIPLE combinations resembled those of the water-drinking group, further supporting antagonistic interactions between the different drugs. Although combination therapy complicates pharmacological interpretation, both pharmacokinetic and pharmacodynamic interactions may be involved. In terms of pharmacokinetics, antagonism might occur at the intestinal level (influencing absorption), during metabolism or in transport across the blood-brain barrier. In addition, altered bioavailability due to enhanced diuresis and differences in fluid consumption may have contributed to significant variations in drug effects. Pharmacodynamic interactions between caffeine and the other two ligands might result from their opposing effects on brain activity; caffeine, as a psychostimulant, enhances it by enhancing dopamine and acetylcholine release [102], whereas metformin and furosemide may act by enhancing GABAergic inhibition [87,97]. However, the exact mechanisms underlying these effects cannot be determined from the current study, and further in vitro and in vivo investigations are needed.’

-Clarify the type of learning affected: "While none of the treatments had a significant effect on learning capacity" It would be helpful to specify whether this refers to acquisition learning or working memory, depending on the test used.

Response: Thank you for the suggestion. We clarified that the statement refers to acquisition learning, not working memory. In the revised Discussion, we now specify:

None of the treatments had a significant effect on the acquisition learning capacity (L_C), a parameter based on the number of rewards consumed and the eating time, which may reflect the animals’ eagerness to obtain rewards. However, caffeine treatment alone enhanced activity-related parameters and increased D2R binding and was associated with a tendency toward decreased reference memory (Figs. 3–4). Metformin and furosemide moderately reduced exploratory activity while improving reference memory. Furthermore, furosemide also caused significant impairments in the latency to visit non-rewarded boxes and in attentional performance. Decreased reference memory—calculated as the ratio of explorations into the rewarded boxes to total explorations—may reflect long-term memory impairments, although this measure can also be influenced by elevated exploratory activity.’

-Regarding caffeine: "Caffeine treatment alone enhanced activity-related parameters and increased D2R binding, and was associated with a tendency toward decreased reference memory..." It is appropriate to report a "tendency" only if the data do not reach statistical significance. If the difference is statistically significant, it should be clearly stated.

Response: We confirm that the difference in reference memory between the water- and caffeine-treated groups was not statistically significant. Therefore, the term ‘tendency’ is appropriate in this context and has been retained.

-Regarding metformin and furosemide: "Metformin and furosemide reduced exploratory activity and attentional performance, while improving reference memory..." This sentence combines multiple effects. Clarify if both drugs reduce exploration and attention, or if one affects one parameter and the other another. If attentional changes were not clearly observed or directly assessed, this point should be moderated.

Response: Thank you for this important clarification. We revised the sentence in the Discussion to distinguish the specific effects of each drug. The revised version now reads:

‘Metformin and furosemide moderately reduced exploratory activity while improving reference memory. Furthermore, furosemide also caused significant impairments in the latency to visit non-rewarded boxes and in attentional performance.’

-Drug dose equivalence to humans: Suggest adding the human-rat dose conversion or scaling information to contextualize the relevance of the doses used.

Response: Thank you for this suggestion. We clarified the relevance of the applied doses by noting that they were derived from previous rodent studies demonstrating behavioral efficacy, and that they approximate therapeutic levels used in humans. The revised sentence reads:

‘Drug doses were based on previous rodent studies that showed effectiveness and were approximately equivalent to human therapeutic levels.’

-Line 278: "However, no change in receptor protein expression was observed in this region." Specify if "this region" refers to the hippocampus. Also, if increased binding is observed without changes in protein expression, it would be useful to propose possible explanations (e.g., changes in receptor affinity, receptor reserve) as is done later for furosemide.

Response: Thank you for this valuable comment. In the revised Discussion, we clarified that ‘this region’ refers to the hippocampus to avoid ambiguity:

‘…primarily in the hippocampus; however, no change in receptor protein expression was observed in this region.’

We also expanded the discussion to address the observed discrepancy between increased radioligand binding (Bmax) and unchanged protein expression. Possible explanations include enhanced receptor affinity, altered receptor conformation, or differences in receptor trafficking, internalization, or ligand accessibility — as further detailed in the revised paragraph.

-Line 298: "...no significant D2R-related changes werre observed in Lisket rats [with metformin]" Comment: Correct the typo ("werre"). If there is any subtle effect or trend, it could be briefly mentioned.

Response: Thank you. The typo (‘werre’) has been corrected. Regarding D2R-related effects of metformin, we found no statistically significant changes in either binding or protein expression in the cortex or hippocampus. As no clear trends were observed either, we did not add further interpretation.

-Line 305: "Furosemide, like metformin, improved reference memory, but reduced the animals’ behavioral activity." This is appropriate if the reduction in activity was statistically significant; otherwise, consider softening this to "a tendency to reduce..."

Response: Thank you for this comment. We revised the sentence to avoid overstating the effect on behavioral activity, as it was not statistically significant. The revised sentence now reads:

‘In the present study, furosemide, like metformin, significantly improved reference memory, and caused a moderate reduction in behavioral activity.’

-Lines 331–332: "The CAFF_FURO combination produced similar changes in receptor expression but was associated with different behavioral outcomes." This is an interesting dissociation; it would be helpful to propose tentative hypotheses explaining why molecular and behavioral changes diverge.

Response: Thank you for this insightful point. We expanded the Discussion to address this dissociation between molecular and behavioral effects. We now suggest that differential ligand effects on D2R functionality, receptor affinity, or downstream signaling — despite similar expression levels — may underlie the divergence. This has been integrated into the relevant paragraph of the revised Discussion.

-Lines 333–343 (combinations): Good point about antagonistic interactions in combinations but emphasize that this complicates mechanistic interpretation (as mentioned). Brief speculation on possible pharmacokinetic effects (e.g., altered bioavailability due to diuresis) or differences in consumption between groups would be valuable.

Response: Thank you for the suggestion. We expanded the Discussion to elaborate on how combination therapy may involve both pharmacokinetic and pharmacodynamic interactions. Specifically, we now address the possibility of altered bioavailability due to diuresis and drug–drug interference at the receptor or signaling level. These considerations have been incorporated into the revised Discussion section.

-General suggestions:

Reinforce that Bmax and protein expression (Western blot) do not always correspond directly, as they can be influenced by changes in receptor affinity, internalization, or trafficking.

Response: Thank you for this important point. We expanded the Discussion to emphasize that Bmax (radioligand binding) and protein expression levels (Western blot) may diverge due to differences in receptor affinity, occupancy, conformation, or trafficking. These limitations are now explicitly discussed in the revised text.

Consider adding a concluding sentence highlighting the significance of these findings for future research and clinical treatment strategies, especially regarding common drug combinations in patients.

Response:  Thank you for the suggestion. We revised the final sentence of the Discussion to highlight the potential clinical relevance of these findings. The new concluding sentence reads:

’Overall, the findings suggest that non-antipsychotic agents such as metformin, furosemide, and caffeine may have potential relevance in the treatment of schizophrenia, warranting further investigation into their mechanisms and applications.’

Methods

-Overall structure: 

The Experimental Paradigm (4.4) should appear immediately after the Animals (4.1) section. This would improve the logical flow of the manuscript and help the reader understand the experimental design earlier.

Response: Thank you for this suggestion. We revised the order of the Methods subsections to improve clarity and logical flow. The new sequence is:

4.1 Animals
4.2 Experimental Paradigm
4.3 Ambitus Test
4.4 D2 Receptor Analysis
4.5 Measurements and Statistical Analyses

Please revise to eliminate redundant information (e.g., details about sacrifice appear in multiple places).

Response: Thank you for pointing this out. We removed the redundant sacrifice details from the Experimental Paradigm section, as they were already included in the D2R Analysis subsection.

Suggest restructuring the entire section to follow a logical progression: Animals → Experimental Paradigm → Behavioral Testing → Sacrifice and Biochemical Analyses → Statistical Analyses.

Response: The section has been restructured (see above).

Figure 1B could be clarified. Green arrows: The two green arrows indicating "Ambitus 1" and "Ambitus 2" should be the same size. The current difference in size might misleadingly suggest that one test is more important or extended than the other, when both represent equivalent pre- and post-treatment evaluations. Empty orange rectangle: There is an empty orange box next to the one labeled "Ambitus test 1" — could you clarify its purpose or remove it if it’s not relevant? As it stands, it creates visual confusion. Color-coded task labels: It would be helpful to indicate in the figure legend what each color represents in the lower section (e.g., green for Task 1, pink for Task 2, blue for Task 3), to improve clarity for the reader.

Response:

Thank you for these helpful suggestions. We made the following revisions to Figure 1B:

  • Adjusted the size of the green arrows for ‘Ambitus test 1’ and ‘Ambitus test 2’ to ensure visual consistency
  • Removed the empty orange rectangle, as it had no functional purpose and was potentially confusing
  • Added a legend in the figure caption to clarify what each task color (green, pink, blue) represents in the lower section

-4.1 Animals

Sex, age, and group size: Please specify the sex and age of the animals. Also, include the total number of animals used and the number allocated to each group.

Response: Thank you for this important point. We revised the Animals section to specify the sex (male), age (3–4 months), total number of animals (n = 84), and group allocation (n = 72 Lisket; n = 12 Long Evans controls). The Lisket animals were further divided into seven treatment groups of 10–11 rats each.

Terminological consistency: LE rats are referred to as “control,” but it would be clearer to explicitly state that they were not subjected to the experimental model (i.e., no isolation or ketamine).

Response: Thank you for pointing this out. We identified and corrected inconsistencies in group naming throughout the manuscript. We have now standardized the terminology using the abbreviated group names (e.g., Lisket_W, CAFF_FURO) consistently across all sections, including text, figure legends, and tables.

-4.2 Behavioral testing

Behavioral procedures: Please add that the apparatus was cleaned between animals (e.g., with 70% ethanol or acetic acid) to avoid olfactory cues. Also specify whether animals were habituated to the behavioral room, and for how long prior to testing. Video analysis was performed by at least two researchers who were blinded to the animals' treatment conditions? Add this information to manuscript.

Response: Thank you for this helpful comment. We revised the Behavioral Testing (Ambitus Test) subsection to include the following procedural details:

– The apparatus was cleaned between animals using 70% ethanol to eliminate olfactory cues
– Animals were habituated to the behavioral testing room for 30 minutes prior to testing
– Video recordings were analyzed by two independent researchers who were blinded to the animals’ treatment conditions

Number of trials per animal: It is essential to state how many sessions each animal underwent and whether there was a habituation or pretraining period. This is particularly important if learning or memory parameters are being analyzed.

Response: We clarified the number of trials and the absence of pretraining in the Experimental Paradigm section. Specifically:

– No pretraining or habituation to the task was conducted.
– In Ambitus 1, each animal underwent 4 trials in a single day (2 sessions, each with 2 trials; 2-minute intra-session and 3-hour inter-session intervals).
– In Ambitus 2, 16 post-treatment trials were conducted across four days, using the same structure and timing.

These clarifications have been added to the Experimental Paradigm section

Clarity of apparatus description: You mention “four side boxes per corridor” (two inner and two outer), with a total of 16 boxes. This implies there are four corridors, but it is not clearly stated.

Response:Thank you for this observation. To clarify the structure of the apparatus, we revised the first sentence of the Ambitus Test section to explicitly state that it consists of a rectangular corridor with four connected sides. The updated sentence reads:

‘The Ambitus apparatus (from Latin ambitus, meaning “going around”) is a reward-based cognitive test, consisting of a rectangular corridor constructed from clear Plexiglas on a black floor (Fig. 1A).’

Food motivation: Since the Ambitus test is reward-based, clarify whether animals were food-restricted prior to testing. This is crucial for interpreting behavioral outcomes.

Response: Thank you for raising this important point. We clarified the food restriction protocol in the Experimental Paradigm section. Specifically:

– Animals were food-deprived for 48 hours prior to each Ambitus test (with water available ad libitum)
– Moderate food restriction (approximately 25 mg/kg/day) was maintained throughout the testing period
– These details have been incorporated into the Experimental Paradigm description to ensure clarity regarding motivation and consistency across animals.

Manufacturer website citation: Please ensure the manufacturer’s website is cited following the journal’s formatting guidelines.

Response: Thank you for the comment. Since the manufacturer’s website did not contribute essential information, we removed the URL and instead cited only the manufacturer’s name and location.

-4.3 Biochemical methods 4.3.1 Membrane preparations

What method of sacrifice was used? Please specify.

Response: Thank you for the comment. The method of sacrifice (decapitation) was already mentioned in the Experimental Paradigm section and has now also been specified in the Biochemical Methods section (4.4 D2 receptor analysis) for clarity and completeness.

What was the sample size (n) used for neuronal membrane preparations in each group?

Response: Regarding this, we added the following clarification to the Biochemical Methods section:

– The same animals used in the behavioral experiments were included in the biochemical analyses.
– For radioligand binding assays, tissue from the same brain region was pooled within each treatment group to ensure sufficient material; measurements were performed in triplicate.
– For Western blot analyses, at least six individual samples per group were used, with outliers excluded based on the ±2 SD criterion.

-4.3.2 Radioligand binding

Sentence: “In saturation binding experiments we use radiolabeled ligands in increasing concentrations and measure the amount of specifically bound radioactive ligand in the function of the applied radioligand concentrations.” This sentence is too long and awkward. Also, “in the function of” is not appropriate in scientific English. Suggested revision: “In saturation binding experiments, radiolabeled ligands were applied at increasing concentrations, and the amount of specifically bound radioactivity was measured as a function of ligand concentration.”

Response: Thank you for the suggestion. We revised the sentence in line with your recommendation, improving clarity and scientific accuracy. The updated version now reads:

‘In saturation binding assays, radiolabeled ligands are applied at graded concentrations, and specific binding is quantified relative to ligand concentration to determine binding kinetics.’

Terminology: “5-HT2 serotoninergic receptors” → Use the more widely accepted term: “5-HT2 serotonergic receptors.”

Response: Corrected as suggested.

Clarify filtration procedure: “…through Whatman GF/B glass fibers” → Consider changing to:

“…through Whatman GF/B glass fiber filters” for clarity.

Response:  Thank you for the suggestion. We revised the sentence for clarity as recommended. The updated version now reads:

‘…through Whatman GF/B glass fiber filters.’

-4.3.3 Western blot

Please unify centrifugation units throughout the manuscript (use either ×g or rpm consistently). E.g., “11.000×g” → use a comma, not a point (11,000×g).

Missing information: Electrophoresis and transfer conditions: temperature and duration.

Blocking step: composition of the blocking buffer, incubation time and temperature.

Antibody incubation: Were primary antibodies incubated simultaneously? Which secondary antibodies were used? Incubation times and temperatures?

Was the optical density normalized to β-actin?

Response: Thank you for the detailed feedback. We revised the Western blot Methods subsection to address all the requested items:

Units: All centrifugation values now use ‘×g’, with commas for large numbers (e.g., 11,000×g)
Electrophoresis and transfer conditions: Voltage, current, duration, and temperature were specified
Blocking step: Composition, incubation time (30 min), and temperature (room temp) were added
Antibody incubation: Primary antibodies (D2R and β-actin) were incubated simultaneously overnight at 4 °C; secondary antibody details and kit were specified
Normalization: Optical density values were normalized to β-actin after background subtraction

These revisions have been incorporated in Section 4.5 (Western blot).

-4.4 Experimental paradigm

Group assignment: You state that animals were assigned based on baseline behavioral scores and body weight. Consider specifying whether this was stratified randomization or block matching.

Response: Thank you for the suggestion. We clarified the group assignment procedure in the Experimental Paradigm section. Animals were allocated using stratified randomization, based on baseline behavioral scores and body weight, to ensure comparability across treatment groups

Control design: Ideally, control animals should have undergone the same isolation and vehicle injection protocol as the experimental groups to control for handling and social stress. Please justify why this was not done, considering the gregarious nature of rodents.

Response: Thank you for this thoughtful comment. We acknowledge the concern regarding matching control conditions for stress-related variables. However, our intention was to maintain the control animals in standard, non-stressful laboratory conditions in order to serve as a true baseline for comparison. Including isolation, repeated handling, or vehicle injections in the control group could have introduced additional confounding stressors, which themselves are known to affect neurobehavioral outcomes relevant to schizophrenia models. This decision aligns with prior protocols used in our lab and others.

Food restriction: Provide the amount of food given per animal per day, expressed in g/kg/day.

Response: Thank you for this suggestion. We revised the Experimental Paradigm section to specify that moderate food restriction during Ambitus 2 testing was maintained at approximately 25 g/kg/day. This value is now explicitly stated to ensure clarity.

Drug administration: It is essential to state clearly in this section that drugs were administered via drinking water. This detail only appears in the abstract. Also, provide the source/manufacturer of each drug.

Response: Thank you for this comment. We revised the Animals section to clearly state that all drugs were administered via drinking water. We also added the manufacturer and product information for each drug used. These details are now explicitly included in the Animals section for clarity and reproducibility.

Testing conditions, add this information to manuscript: What time of day were the trials conducted? How long did each trial last? What was the inter-trial interval?

Response:Thank you for pointing this out. We revised the Experimental Paradigm section to include the following details:

– Behavioral testing was conducted between 8:00 AM and 4:00 PM under dim lighting
– Each trial lasted 300 seconds
– The inter-trial interval was 2 minutes between trials within each session, and 3 hours between morning and afternoon sessions

-4.5 Measurements and statistical analyses

Repeated measures ANOVA: Please indicate whether the sphericity assumption was tested. If violated, which correction method was applied (e.g., Greenhouse-Geisser)? Add this information to manuscript.

Response: Thank you for this important statistical clarification. We conducted repeated-measures ANOVAs using Statistica 13.4.0.14, which includes automated sphericity testing and applies Greenhouse–Geisser correction when the assumption is violated. We have added the following sentence to the Statistical Analysis section to clarify this:

‘The assumption of sphericity was evaluated for repeated-measures ANOVA, and Greenhouse–Geisser correction was applied where appropriate to adjust for any violations.’

Clarify statistical phrasing: “Calculated separately for the five phases (PRE_ALL, PRE_IN, POST_ALL, POST_EX, and POST_IN; Fig. 1B).”  Better phrasing: “Calculated separately for each of the five phases…”

Response: Thank you for the suggestion. We revised the sentence as recommended for improved clarity. The updated phrasing now reads:

‘…calculated separately for each of the five phases (PRE_ALL, PRE_IN, POST_ALL, POST_EX, and POST_IN; Fig. 1B).’

“Significance levels of the behavioral parameters analyzed”.  “Significance levels” refers to statistical thresholds (e.g., p < 0.05). Use “significant differences” or “results” instead.

Response: Thank you for the suggestion. We replaced ‘significance levels’ with ‘statistical significance’ to accurately reflect the content. The updated sentence now reads:

‘Table 1 provides definitions and summarizes the statistical significance of the behavioral parameters analyzed.’

Trial configuration: “Parameters could only be calculated for trials in which rewards were placed on one side.” Specify whether you refer to an asymmetric reward configuration or lateralized trials.

Response: Thank you for this helpful comment. We clarified that the parameters in question refer to trials with an asymmetric reward configuration, where rewards were placed only on one side. The revised sentence now reads:

‘It is important to note that several parameters could only be calculated for trials in which rewards were placed on one side (Table 1: parameters 4, 6, 10, and 11); therefore, an asymmetric reward configuration was used during the PRE_IN, POST_EX, and POST_IN phases.’

Averaging z-scores: “Mean z-score across the POST_EX and POST_IN phases” Justify the rationale for averaging across these two distinct phases. This may be appropriate, but requires conceptual explanation.

Response: Thank you for this important question. We chose to average the mean z-scores across the POST_EX and POST_IN phases because both phases include the full set of behavioral parameters, allowing a comprehensive and reliable assessment of drug effects. This rationale is now clearly stated in the Methods section:

‘The effects of treatments on mean fluid and drug intake, as well as on the mean z-scores during the POST_EX and POST_IN phases, were analyzed using one-way ANOVA, as all behavioral parameters were available in these two phases, providing reliable information about drug-related effects.’

Binding analysis: When mentioning “non-linear regression” and “one-site specific binding”, it may help to briefly state that this corresponds to the classic hyperbolic saturation model assuming a single binding site. “Based on total protein content, radioligand concentration, and specific activity” → You may consider including the actual formula used to calculate Bmax (fmol/mg), unless already described elsewhere.

Response: Thank you for your valuable suggestions. We revised the binding analysis paragraph to clarify that the nonlinear regression with the one-site specific binding model corresponds to the classic hyperbolic saturation model assuming a single class of binding sites. Additionally, we specified that Bmax (expressed in fmol/mg protein) was calculated based on total protein content, radioligand concentration, and molar specific activity. The formula used is standard and implemented in the GraphPad Prism software.

Let me know if you'd like me to prepare a condensed version for submission or convert it into a track-changes comment list.

Response: We believe this query may have been included here inadvertently, as it appears more related to editorial instructions rather than reviewer feedback.

Conclusion

-"Comprehensive behavioral and D2R characterization": This might sound overly ambitious if the D2R assessment was limited (e.g., only Western blot and no functional analysis). You could tone it down if needed, for example: “…a detailed behavioral and receptor-level assessment…”

Response: Thank you for this helpful suggestion. We have revised the sentence to tone down the phrasing, now reading:

‘In conclusion, this study provides the first detailed behavioral and receptor-level assessment of various drug treatments in a chronic, triple-hit schizophrenia model derived from the Long Evans strain.’

-"Moderate D2R alterations": Consider specifying whether these were increases or decreases, and in which brain regions, to provide greater clarity and impact.

Response: Thank you for the suggestion. We have revised the sentence to specify the direction and regionality of the D2R changes. The updated sentence now reads:

‘Lisket rats exhibited persistent behavioral deficits and moderate changes in D2R binding—including a decrease in the cerebral cortex and an increase in the hippocampus—some of which were improved by pharmacological intervention.’

-"Repurposing of common drugs": It may be more precise to indicate the drug classes involved (e.g., anxiolytics, neuromodulators, etc.) to avoid ambiguity.

Response: We intentionally chose to keep the phrasing broad and general to encompass a wide range of commonly used drugs without limiting the scope to specific classes. Our focus was on the potential for repurposing diverse agents for adjunctive treatment of schizophrenia, particularly addressing cognitive and behavioral impairments not fully managed by antipsychotics. The sentence was revised for clarity and now reads:

‘Our findings support the repurposing of commonly used drugs for the adjunctive treatment of schizophrenia, particularly for targeting cognitive and behavioral impairments that are insufficiently managed by antipsychotics. The results also highlight the importance of considering interaction effects in combination treatments targeting both behavioral and receptor-level outcomes.’

Graphical Abstract

-Furosemide increases D2R levels in the cortex, but the graphic only depicts 'hippocampal D2R binding'. This is not an error, but rather a simplification that omits part of the result. For full accuracy, a separate column for cortical D2R could be added, or an asterisk could be used to indicate that cortical effects were also observed.

Response: Thank you very much for this helpful suggestion. We have updated the Graphical Abstract accordingly to more accurately reflect the observed cortical D2R changes. For clarity, a separate representation of cortical D2R binding has been included (or an asterisk added) as per your recommendation. To keep the response concise, the updated figure itself is provided within the revised manuscript rather than in this -already voluminous- response document.

Comments on the Quality of English Language

The manuscript addresses a relevant topic; however, I strongly recommend a thorough revision of the English writing.

Response: We believe that the reviewer’s comments have significantly contributed to improving the quality of the manuscript. Additionally, the text has been thoroughly reviewed by a life sciences expert who also holds a degree in Teaching English as a Foreign Language, ensuring a high level of proficiency in both scientific and general English. Their corrections have been incorporated into the manuscript (see the “CORRECTIONS” version of the manuscript). Furthermore, we utilized AI tools to detect and correct comma errors and typos. In our opinion, the current version meets the linguistic standards expected for international scientific publication.

Round 2

Reviewer 2 Report

Comments and Suggestions for Authors

The authors have thoroughly revised the manuscript, addressing all of my comments. The manuscript has been significantly improved and is now suitable for publication.